# Temporal variation of bacterial community and nutrients in Tibetan glacier snowpack

Yuying Chen[1,4], Keshao Liu[1,4], Yongqin Liu[1,2,4], Trista J. Vick-Majors[3], Feng Wang[1,4], Mukan Ji[2]

[1]State Key Laboratory of Tibetan Plateau Earth System, Resources and Environment (TPESRE), Institute of Tibetan Plateau Research, Chinese Academy of Sciences, Beijing 100101, China
[2]Center for the Pan-third Pole Environment, Lanzhou University, Lanzhou 730000, China
[3]Department of Biological Sciences, Great Lakes Research Center, Michigan Technological University, Houghton, Michigan 49931, United States
[4]University of Chinese Academy of Sciences, Beijing 100049, China

*Correspondence to*: Mukan Ji (jimk@lzu.edu.cn)

**Abstract.** The Tibetan Plateau harbours the largest number of glaciers outside the polar regions, which are the source of several major rivers in Asia. These glaciers are also major sources of nutrients for downstream ecosystems, while there is little data available on the nutrient transformation processes on the glacier surface. Here, we monitored the carbon and nitrogen concentration changes in a snowpit following a snowfall in the Dunde Glacier of the Tibetan Plateau. The association of carbon and nitrogen changes with bacterial community dynamics was investigated in the surface and subsurface snow (depth at 0-15 and 15-30 cm, respectively) during a nine-day period. Our results revealed rapid temporal changes in nitrogen (including nitrate and ammonium) and bacterial communities in both surface and subsurface snow. Nitrate and ammonium concentrations increased from 0.44 to 1.15 mg/L and 0.18 to 0.24 mg/L in the surface snow and decreased from 3.81 to 1.04 mg/L and 0.53 to 0.25 mg/L in the subsurface snow over time. Therefore, we suggest that the surface snow is not nitrogen-limited, while the subsurface snow is associated with nitrogen consumption processes and nitrogen-limited. The nitrate concentration co-varied with bacterial diversity, community structure, and the predicted nitrogen fixation and nitrogen assimilation/denitrification-related genes (*narG*), suggesting nitrogen could mediate bacterial community changes. The nitrogen limitation and enriched denitrification-related genes in subsurface snow suggested stronger environmental and biotic filtering than those in surface snow, which may explain the lower bacterial diversity, more pronounced community temporal changes, and stronger biotic interactions. Collectively, these findings advance our understanding of bacterial community variations and bacterial interactions after snow deposition and provide a possible biological explanation for nitrogen dynamics in snow.

# 1 Introduction

The Tibetan Plateau is the world's third-largest ice reservoir, after those in Antarctica and Greenland (Qiu, 2012). These glaciers are the source of several large rivers in Asia, such as the Yellow, Yangtze, Mekong, Salween, Brahmaputra, and Indus rivers (Immerzeel et al., 2010). Glaciers are major sources of nutrients (carbon and nitrogen) for the downstream ecosystems (Singer et al., 2012; Hood et al., 2015; Liu et al., 2021). It has been estimated that 80 gigagram of dissolved organic carbon and 27-43 gigagram of nitrogen are exported from the Greenland Ice Sheet (Bhatia et al., 2013; Wadham et al., 2016). These

nutrients are subjected to complex accumulation and transformation processes in the glacier snow before being released into downstream ecosystems, and microorganisms are the drivers of these processes (Anesio and Laybourn-Parry, 2012; Hell et al., 2013; Hodson et al., 2008). Several studies on snowpacks revealed vital knowledge of the nutrient and microbial community dynamics in the Arctic (Hell et al., 2013; Larose et al., 2013a; Larose et al., 2013b; Maccario et al., 2014; Maccario et al., 2019), Antarctic (Antony et al., 2016), and Alps (Lazzaro et al., 2015). However, such knowledge is rarely available in the

Tibetan Plateau, constraining our understanding of the nutrient accumulation, transformation, and release processes, which is urgently needed under the enhanced warming and glacier retreat in the Tibetan Plateau.

Autochthonous (microbial origin) and allochthonous (wet and dry atmospheric depositions) are the major sources of nutrients on supraglacial snow, and the contribution of allochthonous sources was much greater in Arctic glaciers (Larose et al., 2013a). Microorganisms are highly involved in the transformation of both autochthonous and allochthonous nutrients.

Several studies investigated the dynamics of nutrient and bacterial changes in supraglacial snow during the ablation period. Larose et al. (2013a) revealed that the form of nitrogen varied as a function of time in supraglacial snow during a two-month field study in Svalbard, and fluctuations in microbial community structure were reported with the relative abundance of fungi and bacteria (such as Bacteroidetes and Proteobacteria) increased and decreased, respectively. Seasonal shifts in snowpack bacterial communities were reported in the mountain snow in Japan, where rapid microbial growth was observed with

increasing snow temperature and meltwater content (Segawa et al., 2005). However, the results of these studies are likely the consequence of several precipitation events due to the long study period. During precipitation, a new snow layer forms above the previous ones, which is responsible for the stratified snowpack structure. These different snow layers have distinct physical and chemical characteristics and their age also differs substantially (Lazzaro et al., 2015). Thus, while the microbial process across the aged snowpack can be complex, focusing on supraglacial snow from a single snowfall event could provide unique

insights into the bacterial and nutrient dynamics. For instance, Hell et al. (2013) reported bacterial community structure changes during the ablation period across five days in the high Arctic, but the bacterial and nutrient dynamics during the snow accumulation period remain elusive.

Surface and subsurface snow typically harbour distinct bacterial community structures (Xiang et al., 2009; Møller et al., 2013; Carey et al., 2016). For example, algae (chloroplasts), Proteobacteria, Bacteroidetes, and Cyanobacteria were more

abundant in surface snow, while Firmicutes and Fusobacteria were more abundant in the deeper snow layer (Møller et al., 2013). A previous study had proposed that nitrogen availability could also be a driver of microbial community structure and

function in snow (Larose et al., 2013b), where the $NO_3^-$ and $NH_4^+$ concentrations drove the community composition in Ny-Ålesund snowpack. A dissolved inorganic nitrogen addition experiment also showed a clear community response with the bacterial abundance elevated and genera richness declined in the final time point compared to the initial time point, suggesting

potential specialization of heterotrophic communities(Holland et al., 2020).

  Differences in physicochemical conditions can also indirectly influence bacterial community structure through impacts on the types of biotic interactions (Friedman and Gore, 2017; Khan et al., 2018; Bergk Pinto et al., 2019). For example, the addition of organic carbon shifted bacterial interactions from collaboration to competition in Arctic snow (Bergk Pinto et al., 2019), with complex organic carbon degradation and mineralization requiring intensive microbial collaborations (Krug et al.,

2020), which are particularly important for oligotrophic environments, such as glaciers. Collaboration is also known to be essential to biological processes such as ammonia oxidation and denitrification, in which various organisms carry out different steps of these processes (Henry et al., 2005; Madsen, 2011; Yuan et al., 2021). These changes in interactions and network complexity can favour or disadvantage certain bacterial groups, thereby changing the bacterial community structure (i.e., biofiltering).

Several studies have investigated the nutrient and bacterial community changes in supraglacial snow across the winter (Brooks et al., 1998; Liu et al., 2006), but the bacterial and nutrient dynamics of freshly fallen snow have been largely overlooked. These short temporal changes will influence the following post-depositional processes after it is buried by the next snowfall, and will ultimately determine the physicochemical properties of the stratified snow in the following year. In the present study, we investigated the bacterial community and snow physiochemical property changes in the surface and

subsurface supraglacial snow during a nine-day period after a single snowfall event at the Dunde Glacier on the northeast of the Tibetan Plateau. We aimed to answer the following key questions: 1) do the bacterial community and nutrient changes in a short temporal scale, 2) do the bacterial communities in different snow layers exhibit similar community temporal changes, and 3) are the temporal changes in the surface and subsurface snow related to environmental filtering, biotic interactions, or both?

**2 Materials and methods**

**2.1 Site description and sample collection**

Snow samples were collected from the ablation zone at the Dunde glacier (38°06′N, 96°24′E, 5325 m above the sea-level), during October and November, 2016 (Supplementary Fig. S1). Dunde glacier is located in the Qilian mountain region on the north-eastern Tibetan Plateau, and it is continuously monitored by the Institute of Tibetan Plateau Research, Chinese Academic

of Sciences. No supraglacial snow was observed on the glacier surface on the 10th of October when first arrived at the camp. Snowfall started on the 18th and ended on the 23rd of October. Sampling was conducted over a nine-day period after the snowfall stopped on a flat 5 m × 3 m small area to reduce the impact of sample heterogeneity due to spatial variations. Snow samples were collected on the 24th, 25th, 26th, 27th, and 29th of October, and the 2nd November (which are referred to as day 1, 2, 3, 4,

6, and 9) until the next snowfall started. This enabled us to monitor the succession of bacterial communities and the chemical changes in snow through time after deposition. The ambient air temperature at the sampling period averaged -8 °C (data available through the European Centre for Medium-Range Weather Forecasts, Supplementary Fig. S2), and no snow melting was observed over the nine day period.

On each day, three snow pits were randomly dug within the 5 m × 3 m area and any two snow pits were 30-50 cm apart. Each snow pit was approximately 30 cm deep, and the snow was further divided equally into the surface and subsurface layers (approximately 15 cm deep for each layer) to get enough snow for DNA extraction, according to Carey et al. (2016). For each snow pit, the top 1 cm in contact with the air was removed using a sterile spoon to avoid contamination, and then surface and subsurface snow were collected using a sterilized Teflon shovel into 3 L sterile sampling bags separately. Approximately 100 mL were used for physicochemical analyses, whereas the rest was used for DNA extraction. A total of 36 samples were collected. Tyvek bodysuits and latex gloves were worn during the entire sampling process to minimize the potential for contamination, and gloves were worn during all subsequent handling of samples. Samples were kept frozen during the transportation to the laboratory and stored at -20 °C until analysis.

## 2.2 Environmental characterization of snow

The 100 mL snow sample allocated for physicochemical analysis was melted at room temperature for 3 hours before being analyzed. For dissolved organic carbon (DOC) and major ions measurements, 100 mL of snow meltwater was syringe-filtered through a 0.45 μm polytetrafluoroethylene (PTFE) membrane filter (Macherey–Nagel) into 20-mL glass bottles. The membrane was pre-treated with 1% HCl, deionized water rinsed, and 450 °C > 3 h combusted to remove any potential carbon and nitrogen on the membrane, and the initial 10 mL of the filtrate was discarded before collecting the sample for analysis to eliminate any residual compound on the membrane. The DOC concentrations were measured with a TOC-VCPH analyzer (Shimadzu Corp., Japan). Major ions ($NH_4^+$, $NO_3^-$, $Na^+$, $K^+$, and $SO_4^{2-}$) were analyzed using a Thermo-Fisher ion chromatography system 900 as described previously (Rice et al., 2012). The precision and accuracy of the TOC-VCPH analyzer were both < 3% and the limit of detection was 0.05 mg $L^{-1}$. The precision and accuracy of the ion chromatography system 900 were < 5% and 0.1 mg $L^{-1}$, and the limit of detection was 0.01 mg $L^{-1}$ (Supplementary Fig. S3).

## 2.3 DNA extraction

For assessing the bacterial community composition, snow samples (3 L) were melted at 4 °C overnight and filtered onto a sterile 0.22 μm polycarbonate membrane (Millipore, USA) with a vacuum pump (Ntengwe 2005). Bacterial community DNA was extracted from the biomass retained onto the filters using a Fast DNA®SPIN Kit for Soil (MP Biomedicals, Santa Ana, CA, USA) according to the manufacturer's instructions. DNA extraction with no sample added was performed in parallel and used as a negative control.

The raw DNA was checked by electrophoresis in 1% (w/v) agarose gel, and purified from the gel using an Agarose Gel DNA purification kit (TaKaRa, Japan). The concentration and purity of the DNA extracts were measured using a NanoDrop

1000 spectrophotometer (Thermo-Scientific, Wilmington, DE, USA). The extracted DNA was stored at -80 °C until amplification.

## 2.4 Bacterial 16S rRNA amplification and Illumina MiSeq sequencing

In total, 36 DNA samples and one negative control were subjected to amplicon sequencing. Universal primers 515F (5'-GTGCCAGCMGCCGCGGTAA-3') and 806R (5'-GGACTACHVGGGTWTCTAAT-3') (Caporaso et al., 2012), with 12 nt unique barcodes, were used to amplify the V4 hyper-variable regions of the bacterial 16S rRNA gene. Polymerase chain reaction (PCR) was performed under the following conditions: 94°C for 5 minutes, 30 cycles of 94°C for 30 seconds, 52°C for 30 seconds, 72°C for 30 seconds; followed by a final cycle of 10 minutes at 72°C. Each PCR reaction contained 12.5 μL 2x Premix Taq DNA polymerase (Takara Biotechnology, Dalian Co. Ltd., China), 1 μL each primer (0.4 μM final concentration), and 8.5 μL nuclease-free water, 2 μL DNA template (20 ng μL$^{-1}$) or 2 μL sterile water for the PCR negative controls. PCR products were confirmed using agarose gel electrophoresis, and no PCR band was detected in the PCR negative controls. To minimise PCR batch-to-batch variations and maximize the quantity of PCR product, triplicate PCR reactions were performed for each sample, and PCR products were pooled for purification using the OMEGA Gel Extraction Kit (Omega Bio-Tek, Norcross, GA, USA) following electrophoresis. PCR products from different samples were pooled in equal molar amounts, and then used for $2 \times 250$ bp paired-ends sequencing on a MiSeq machine (Illumina, San Diego, CA).

## 2.5 Processing of Illumina sequencing data

MiSeq sequence data were processed using the QIIME 2 pipeline version 2018.8 (Bolyen et al., 2018), following the recommended procedures (https://docs.qiime2.org/2018.8/tutorials/) and using the plugin demux to visualize interactive quality diagrams and check read quality. Plugin DADA2 (Callahan et al., 2016) was applied to remove primers, truncate poor-quality bases, conduct de-replication, identify chimeras, and merge paired-end reads. Commands included in the feature table (McDonald et al., 2012) generated the summary statistics of sequences related to the samples. Further, we trained a Naïve Bayes Classifier with the feature-classifier plugin using the 16S rRNA gene database at 99% similarity of the SILVA 132 QIIME release and based on the 515F/806R primer pair as used for the PCR. Finally, the taxa plugin was used to filter mitochondrial and chloroplast sequences, as well as to generate absolute read count tables of all taxa for each sample. Data were analyzed at the level of amplicon sequence variant (ASV), where ASVs are delineated by 100% sequence identity (Callahan et al., 2017).

After removing singletons, a total of 1,685,186 high-quality reads were obtained, representing 9178 ASVs. Before statistical analysis, the dataset was rarefied to 45,000 reads per sample, which is the lowest read count among samples. Rarefaction curves reached an asymptote before the subsampling, which confirmed that this depth was sufficient to detect the diversity present (Supplementary Fig. S4).

## 2.6 Network analysis

The ASV-ASV associations within the surface and subsurface bacterial communities were explored using Molecular Ecological Network Analyses Pipeline (http://129.15.40.240/mena/) (Deng et al., 2012). The ASVs that occurred in at least 50% of the samples from the surface or subsurface group were selected to construct the network. Spearman's rank correlation coefficient ($\rho$) was calculated to reflect the strength of association between species. The false discovery rates (Q-values) were calculated from the observed P-value distribution. The resulting correlation matrix was analyzed with the Random Matrix Theory (RMT)-based network approach to determine the correlation threshold for network construction, and the same threshold was used for both the surface and subsurface network, so the topological properties of the surface and subsurface networks are comparable.

## 2.7 Statistical analysis

Shannon-Wiener and Chao1 indices, which were used to estimate the species richness in the snow community, were calculated using the "*diversity*" function in the R package "*vegan*" (Oksanen et al., 2010). Functional profiling of bacterial taxa was carried out using the package "*Tax4Fun2*" in R (Wemheuer et al., 2020). While the application of functional profiles predicted from 16S rRNA gene-based community composition data is limited by the functional information available in databases, we present these data as one possible interpretation of the patterns detected, and note that the "*Tax4Fun2*" package performed well compared to older widely used programs (Wemheuer et al., 2020). The pairwise Wilcoxon rank-sum test was used to compare the depth-horizon differences in environmental variables, alpha-diversity, and the relative abundance of taxonomic groups at the phylum level. Linear regression modeling was implemented in R using the "*lm*" function to estimate the trend of environmental characteristics, alpha diversity, and microbial community composition changes. Multiple linear regression analysis was performed to determine the contribution and significance of the environmental characteristics to the alpha diversity using the "*lm*" function in R. We use the stepwise Akaike information criterion (AIC) method for variable selection by the "*step*" function in R. The best model was chosen based on the lowest AIC value (Wagenmakers and Farrell, 2004). The bacterial community structure was subjected to principal coordinate analysis (PCoA) carried out using the "*pcoa*" function of the "*ape*" package in R. The significance of dissimilarity of community composition among samples was tested using permutational multivariate analysis of variance (PERMANOVA) based on Bray-Curtis distance metrics with the "*adonis*" function in the R package "*vegan*" (Oksanen et al., 2010). Test results with $P < 0.05$ were considered statistically significant. Mantel test based on Spearman's rank correlations was performed using the bacterial dissimilarity and environmental dissimilarity matrix, calculated based on the Bray-Curtis distance metrics and Euclidean distance metrics in the "*vegan*" R package, respectively. The normalized stochasticity ratio (NST) based on the Bray–Curtis dissimilarity was calculated using the "*NST*" package in R to estimate the determinacy and stochasticity of the bacterial assembly processes with high accuracy and precision (Ning et al., 2019). The NST index used 50% as the boundary point between more deterministic (<50%) and

more stochastic (>50%) assembly processes. All environmental variables were normalized before the calculation. All statistical analyses were executed in R version 3.4.3 (R Core Team, 2017).

## 3 Results

### 3.1 Environmental characteristics of the snowpack

The concentrations of $NO_3^-$ and $NH_4^+$ ranged from 0.44 to 5.09 mg $L^{-1}$ and 0.17 to 0.62 mg $L^{-1}$, respectively (Fig. 1a, Supplementary Table S1), and they were both significantly higher in the subsurface than in the surface snow (Wilcoxon rank-sum test; all $P < 0.001$, Fig. 1a). $K^+$ and $SO_4^{2-}$ ions in the subsurface snow were also significantly higher (0.29 ± 0.13 and 6.09 ± 3.18 mg $L^{-1}$, respectively) than those in the surface snow (0.12 ± 0.08 and 3.71 ± 1.64 mg $L^{-1}$; Wilcoxon rank-sum test; $P < 0.001$, and $P = 0.015$, respectively). The concentrations of DOC ranged from 0.46 to 5.89 mg $L^{-1}$ and exhibited no significant difference between the surface and subsurface snow (Wilcoxon rank-sum test; $P = 0.310$). The concentrations of $Na^+$ ion ranged from 0.35 to 7.34 mg $L^{-1}$ and was no significant difference between the surface and subsurface snow (Wilcoxon rank-sum test; $P = 0.079$). The concentration of $NO_3^-$ and $NH_4^+$ ions in the surface snow exhibited a weak, but significantly positive association with time ($F_{1,16} = 5.97$, $P = 0.027$, $R^2 = 0.27$ and $F_{1,16} = 8.58$, $P = 0.010$, $R^2 = 0.35$, respectively, Fig. 1b). On the other hand, stronger negative associations were found between inorganic nitrogen and time in the subsurface snow ($F_{1,16} = 40.66$, $P < 0.001$, $R^2 = 0.72$ and $F_{1,16} = 50.74$, $P < 0.001$, $R^2 = 0.76$, respectively). Other environmental parameters exhibited no significant changes with time.

### 3.2 Diversity and composition of bacterial community from the snowpack

The surface and subsurface snow were both dominated by Alphaproteobacteria, Actinobacteria, Cyanobacteria, Gammaproteobacteria, Bacteroidetes, Firmicutes, Chloroflexi, Gemmatimonadetes, Planctomycetes, Acidobacteria, Deltaproteobacteria, and Deinococcus-Thermus (Fig. 2). The relative abundance of most of these phyla was not significantly different in the two snow layers, except the Gemmatimonadetes, Planctomycetes, and Acidobacteria, which were significantly more abundant in the surface layer than in the subsurface layer (all $P < 0.05$, Wilcoxon rank-sum test; Supplementary Fig. S5). In the surface layer, weak, but significant negative trends were observed between the relative abundances and ASV number of Alphaproteobacteria, Gammaproteobacteria and Firmicutes, and time ($F_{1,16} = 6.97$, $P = 0.018$, $R^2 = 0.30$; $F_{1,16} = 23.8$, $P < 0.001$, $R^2 = 0.60$, and $F_{1,16} = 22.28$, $P < 0.001$, $R^2 = 0.58$ in relative abundance; $F_{1,16} = 7.56$, $P = 0.014$, $R^2 = 0.32$; $F_{1,16} = 27.12$, $P < 0.001$, $R^2 = 0.63$, and $F_{1,16} = 16.68$, $P = 0.001$, $R^2 = 0.51$ in ASV number, respectively), while weak positive correlations were observed between the relative abundances and ASV number of Cyanobacteria and Deinococcus-Thermus, and time ($F_{1,16} = 6.94$, $P = 0.018$, $R^2 = 0.30$ and $F_{1,16} = 13.10$, $P = 0.002$, $R^2 = 0.45$ in relative abundance; $F_{1,16} = 3.42$, $P = 0.083$, $R^2 = 0.18$ and $F_{1,16} = 4.07$, $P = 0.061$, $R^2 = 0.20$ in ASV number, respectively; Supplementary Fig. S6 and S7). Relative to the surface snow, the subsurface layer had stronger negative correlation between the relative abundance and ASV number of Alphaproteobacteria and Firmicutes, and time ($F_{1,16} = 15.17$, $P = 0.001$, $R^2 = 0.49$ and $F_{1,16} = 15.43$, $P = 0.001$, $R^2 = 0.49$ in relative abundance;

$F_{1,16} = 18.98$, $P = 0.083$, $R^2 = 0.54$ and $F_{1,16} = 15.17$, $P = 0.001$, $R^2 = 0.53$ in ASV number, respectively, Supplementary Fig. S6 and S7), while weak correlations were observed between the relative abundance and ASV number of Cyanobacteria and Chloroflexi, and time ($F_{1,16} = 5.62$, $P = 0.031$, $R^2 = 0.26$ and $F_{1,16} = 12.81$, $P = 0.003$, $R^2 = 0.44$ in relative abundance; $F_{1,16} = 5.34$, $P = 0.034$, $R^2 = 0.25$ and $F_{1,16} = 14.49$, $P = 0.002$, $R^2 = 0.47$ in ASV number, respectively).

The bacterial Shannon and Chao1 indices in the surface snow were $5.61 \pm 0.39$ and $744 \pm 199$, respectively, and were not significantly different from those in the subsurface layer ($5.52 \pm 0.68$ and $705 \pm 269$, respectively) ($P = 0.81$ and $0.57$, respectively) (Fig. 3a). In the surface snow, the Shannon and Chao1 indices were similar across the nine days ($F_{1,16} = 0.37$, $P = 0.553$, $R^2 = 0.02$ and $F_{1,16} = 0.01$, $P = 0.939$, $R^2 = 0.001$, respectively; Fig. 3b). Besides, weak positive associations of Shannon and Chao1 indices with the DOC and sodium ions were detected ($F_{1,16} = 4.90$, $P = 0.042$, $R^2 = 0.23$ and $F_{1,16} = 4.91$, $P = 0.042$, $R^2 = 0.24$, respectively; Fig. 4a, b). In contrast, although weak, significant negative correlations were observed in both Shannon and Chao1 indices with time in the subsurface snow ($F_{1,16} = 12.33$, $P = 0.003$, $R^2 = 0.44$ and $F_{1,16} = 8.73$, $P = 0.009$, $R^2 = 0.35$, respectively). Weak, but significant positive associations of Shannon and Chao1 indices with the concentrations of $NO_3^-$ and $NH_4^+$ were detected (Shannon diversity: $F_{1,16} = 9.13$, $P = 0.008$, $R^2 = 0.36$ and $F_{1,16} = 5.17$, $P = 0.037$, $R^2 = 0.24$, respectively; Chao1 index: $F_{1,16} = 8.60$, $P = 0.009$, $R^2 = 0.36$ and $F_{1,16} = 5.32$, $P = 0.035$, $R^2 = 0.25$, respectively; Fig. 4c, d). This is consistent with the multiple linear regression results, which consistently identified the concentrations of $NO_3^-$ and $NH_4^+$ as the significant determinants of bacterial Shannon diversity in the subsurface layer (Supplementary Table S2).

### 3.3 Bacterial community structure and functional genes

The bacterial community structure at the ASV level significantly differed in the surface and subsurface snow (PERMANOVA, $F = 2.78$, $P < 0.001$, Fig. 5a), as well as among the different sampling times (PERMANOVA, $F = 3.31$, $P < 0.001$ and $F = 2.17$, $P < 0.001$, respectively). Additionally, a significant interactive effect was detected between the depth and time (PERMANOVA, $F = 2.68$, $P < 0.001$), indicating that the depth influenced the temporal pattern of bacterial community structure changes. Specifically, only the second principal coordinate (PCoA2) values of the surface snow significantly varied with time ($F_{1,16} = 141.8$, $P < 0.001$, $R^2 = 0.89$, Fig. 5b), while the PCoA1 values of the surface snow did not ($F_{1,16} = 0.04$, $P = 0.840$, $R^2 = 0.003$, Fig. 5b). Furthermore, PCoA1 and PCoA2 of the surface snow exhibited no significant correlation with the measured environmental factors (all $P > 0.05$, Supplementary Fig. S8 and S9). In comparison, both PCoA1 and PCoA2 values of the subsurface, albeit weakly, co-varied with time ($F_{1,16} = 6.35$, $P = 0.023$, $R^2 = 0.28$ and $F_{1,16} = 8.38$, $P = 0.011$, $R^2 = 0.34$, respectively, Fig. 5b), while the PCoA2 also demonstrated significant association with nitrate, ammonium, potassium, sulfate, and DOC concentrations (all $P < 0.05$, Supplementary Fig. S9).

Normalized stochasticity ratio (NST) was used to examine the relative contributions of stochasticity and determinism in shaping bacterial communities. The average NST values were 74% and 46% in the surface and subsurface snow layers, and the contribution of stochasticity was significantly higher in the surface than in the subsurface layers ($P < 0.001$; Supplementary Fig. S10).

Mantel tests were performed to evaluate the effects of environmental factors on bacterial community structure for each layer. No significant correlation was identified between the measured environmental factors and the bacterial community structure in the surface snow. However, weak positive associations were apparent in the subsurface snow with the concentrations of $NO_3^-$ and $NH_4^+$ ($P = 0.005$ and $0.01$, respectively) (Table 1). The relative abundance of nitrogen-cycling associated functional genes was predicted in the surface and subsurface snow. The relative abundance of nitrogen-fixation marker gene (*nifH*) positively associated with time in the surface layer, while no clear pattern was observed in the subsurface layer ($F_{1,16} = 7.76$, $P = 0.013$, $R^2 = 0.33$ and $F_{1,16} = 0.57$, $P = 0.461$, $R^2 = 0.01$, respectively, Supplementary Fig. S11). The relative abundance of *narG* gene, which is involved in the nitrate reduction and denitrification process, exhibited negative and positive associations with time in the surface and subsurface, respectively ($F_{1,16} = 4.69$, $P = 0.046$, $R^2 = 0.23$ and $F_{1,16} = 11.24$, $P = 0.004$, $R^2 = 0.41$, respectively). The *nirK* gene, which is also involved in the denitrification process, decreased with time in the surface layer, while no significant change was detected in the subsurface layer ($F_{1,16} = 10.39$, $P = 0.005$, $R^2 = 0.39$ and $F_{1,16} = 1.98$, $P = 0.179$, $R^2 = 0.05$, respectively).

### 3.4 Interspecies interactions at the surface and subsurface layers

Co-occurrence networks were constructed for the surface and subsurface bacterial communities to infer the biotic interactions among species (Fig. 6). The surface network comprised a higher number of nodes (each indicating one ASV, nodes number = 197), but a lower number of edges (each indicating a significant association between two ASVs, edges number = 436) than the subsurface network (nodes number = 140 and edges number = 523, Table 2). The network in the subsurface snow, relative to surface snow, demonstrated a higher number of edges per node (3.73 and 2.21, respectively), higher average connectivity (avgK, 7.57 and 4.43, respectively), and lower average path distance (GD, 4.72 and 5.51, respectively), which indicate a substantially more complex network topology. Both networks were dominated by positive (co-presence) relationships, and the subsurface network exhibited a higher positive-to-total interaction ratio (95%) than the surface network (83%).

Modularity, average clustering coefficient (avgCC), and graph density of the surface and subsurface bacterial community networks were all higher than those of random networks (Supplementary Table S3), indicating that snowpack bacterial networks showed non-randomly assemblage and exhibited modular structures. The subsurface networks showed higher values of avgCC (0.39), transitivity (0.49), and connectedness (0.86) than the surface bacterial community network (0.31, 0.45, 0.71, respectively), indicating a greater degree of connectivity (Table 2).

## 4 Discussion

### 4.1 Rapid shifts of bacterial community structure across a short temporal scale

The surface and subsurface snow were both dominated by Alphaproteobacteria, Actinobacteria, Cyanobacteria, Gammaproteobacteria, and Bacteroidetes (Fig. 2). Despite differences in sampling season, the bacterial taxa detected were consistent with previous studies on snow in the Arctic and Antarctic (Larose et al., 2010; Carpenter et al., 2000; Amato et al., 2007; Lopatina et al., 2013; Møller et al., 2013). Bacterial richness and diversity exhibited little change throughout the nine days in the surface snow layer, while they exhibited a reduction trend in the subsurface snow layer (Fig. 3b). This indicates

that the microbiome in the subsurface snow may be subjected to greater environmental filtering than those in the surface snow (Xiang et al., 2009). Among all environmental factors measured, nitrate and ammonium were the only measured environmental factors that changed across the nine days. The nitrate and ammonium concentrations in the subsurface snow both exhibited an $R^2$ value of greater than 0.7 and reduced with time, therefore indicating a consumption process (Fig. 1b). Despite the $R^2$ value being weak, both nitrate and ammonium concentrations co-varied with bacteria richness and diversity in subsurface snow,

which was not observed in the surface snow (Fig. 4). Furthermore, multiple linear regression analyses also identified nitrate and ammonium to be the dominant driver of bacteria Shannon diversity in the subsurface snow (Supplementary Table S2). Thus, these results suggest that nitrate and ammonium could play a more important role in influencing bacterial diversity in subsurface snow than that in surface snow. Nitrogen is an essential nutrient for microbial growth and plays important role in controlling microbial diversity and ecosystem productivity (Vitousek et al., 2002; Xia et al., 2008; Sun et al., 2014). The

positive associations between nitrogen concentration and alpha diversity indices have been typically inferred as nitrogen limitation (Telling et al., 2011). Thus, these results hint that nitrogen limitation could occur in subsurface snow and influence bacteria diversity. In comparison, the surface layer is unlikely to be subjected to nitrogen-limitation and the nitrogen in the surface snow slightly increased. This is consistent with previous studies in the Greenland ice sheet, where nitrate additions to surface ice did not alter the cryoconite community cell abundance and 16S rRNA gene-based community composition

(Cameron et al., 2017).

The bacterial community structure also exhibited temporal changes in the subsurface layer. Furthermore, associations between nitrogen and the microbial community structure were observed to a certain degree (Table 1 and Fig. 5), again indicating some level of environmental filtering (Kim et al., 2016). This is consistent with the finding in the Arctic that nitrogen influences snow bacterial community composition via regulating algae metabolism (Lutz et al., 2017). This is also consistent

with the higher contribution of deterministic processes in the subsurface layer than in the surface layer (Supplementary Fig. S10). Deterministic processes could be due to environmental filtering or biotic interactions, whereas stochastic processes include dispersal limitation, community drift, and speciation (Stegen et al., 2012). The surface layer could receive nitrogen input through aeolian deposition processes (Björkman et al., 2014), whereas the subsurface snow could only receive limited external microbial and nutrient input through supraglacial meltwater. The latter could be particularly limited during the glacier

deposition period when the glacier surface temperature is below zero degrees (Fig. S2).

Our results suggest that both bacteria and snow physiochemical properties experience changes across the nine days during the snow deposition period in the Tibetan glacier investigated here, and those changes were more stronger in the subsurface layer than in the surface layer. Traditionally, supraglacial snow is recognized as a cold oligotrophic environment with a very slow metabolism rate (Quesada and Vincent, 2012; Marshall and Chalmers, 1997), but increasing evidence has suggested that

bacterial community changes can occur on a short temporal scale. For example, Hell et al. (2013) reported changes in the dominant bacterial phylum Proteobacteria across five days and active bacterial metabolism has been observed in the Greenland Ice Sheet supraglacial ice (Nicholes et al., 2019). In addition, active bacteria affiliated with Proteobacteria have been identified in the Antarctic (Lopatina et al., 2013) and Arctic (Holland et al., 2020) snow at temperatures below zero degrees, therefore supporting the present study that bacterial community changes in nine days could be possible. This indicates that supraglacial

snow can harbour an active bacterial community, which in turn can have an impact in nutrient transformation.

## 4.2 Distinct nitrogen-transformation processes in surface and subsurface snow

Both ammonium and nitrate concentrations showed a weak increasing trend with time in the surface snow (Fig. 1). The weak increase in ammonium could be explained by biogenic emissions due to local plant and animal sources (Filippa et al., 2010), while the increase in nitrate has been largely attributed to atmospheric deposition (Björkman et al., 2014). Nitrogen deposition occurs at a rate of 282 kg N $km^{-2}$ $yr^{-1}$ in the region of our investigation (Lü and Tian, 2007), which equals to 0.19 mg N for the 0.5 m $\times$ 0.5 m area sampled each day (assuming nitrogen deposition occurred evenly across the year). Another potential source of nitrogen input could be nitrogen fixation process (Telling et al., 2011). Bacteria are the only microorganisms that are capable of fixing atmospheric nitrogen (Bernhard, 2010). Potential nitrogen input from microbial processes is supported by the increase in the nitrogen-fixing Cyanobacteria (Supplementary Fig. S6) and *nifH* gene (Supplementary Fig. S11). Cyanobacteria are known as free-living phototrophs capable of nitrogen fixation, especially in extreme environments (Chrismas et al., 2018; Makhalanyane et al., 2015; Levy-Booth et al., 2014). For example, Cyanobacteria were found as the main group of potential nitrogen fixers determined by quantitative PCR with three sets of specific *nifH* primers on the surface of the Greenland Ice Sheet (Telling et al., 2012). The nitrogen fixation rate was not quantified in the present study, but the present study suggests that microbial nitrogen fixation could be an overlooked source of nitrogen in Tibetan glacier snow. Further transcriptomic and nitrogen-isotope analyses may provide additional evidence on the microbial activity in nitrogen fixation.

In contrast with the surface layer, nitrogen concentrations (nitrate and ammonium) significantly decreased in the subsurface snow with time (Fig. 1). A possible explanation for this might be the microbial utilization and photochemical degradation of nitrogen compounds (Björkman et al., 2014). The microbial processes, i.e. nitrate reduction and denitrification process, are evidenced by the increase of *narG* gene (Supplementary Fig. S11) (Telling et al., 2011; Zhang et al., 2020). Alternatively, microorganisms may carry out assimilatory nitrate reduction, which is used to incorporate nitrogen into biomolecules (Larose et al., 2013a; Richardson and Watmough, 1999). The assimilatory process is performed by a range of microorganisms including bacteria, algae, yeasts, and fungi (Huth and Liebs, 1988). Thus, further studies on eukaryotes, including algae, may provide a full understanding of the nitrogen consumption mechanisms in subsurface snow. The denitrification process converts nitrate to $N_2$ and generates nitrite, nitric oxide (NO), and nitrous oxide ($N_2O$) intermediates (Kuypers et al., 2018). A previous study detected microbial specific phylogenetic probes that targeted genera whose members are able to carry out denitrification reactions such as Roseomonas in a snowpack of Spitsbergen Island of Svalbard, Norway (Larose et al., 2013a). Amoroso et al. (2010) also proposed that denitrification can explain the microbial isotopic signature observed in winter snow at Ny-Alesund. Although the oxygen level in the subsurface snow was not measured, the occurrence of anaerobic denitrification reactions in subsurface snow has been reported in Arctic snowpacks (Larose et al., 2013a). Lastly, photochemical degradation of nitrogen compounds is the most well-known nitrogen degradation pathway, and the release of both NO and $NO_x$ by $NO_3^-$ photolysis on natural snow has been reported in European High Arctic snowpack (Amoroso et al., 2010; Beine et al., 2003). In a snow reactive nitrogen oxides ($NO_y$) survey in Greenland, $NO_y$ flux was reported to exit snow in 52 out of 112 measurements (Dibb

355   et al., 1998). Further metatranscriptomic analyses targeting the genes associated with nitrogen cycling are required to confirm the distinct nitrogen transformation processes between the surface and subsurface layers.

### 4.3 Subsurface snow exhibits greater complexity in biotic interactions

Biotic interactions can explain a substantial proportion of the community structure variations (Hacquard et al., 2015; Dang and Lovell, 2016). Our results indicated that the subsurface community network was more complex as evidenced by the higher average connectivity and a shorter path length (GD), compared to the surface community network (Table 2). This is likely due to the enhanced environmental filtering, as has been observed in other systems subjected to environmental stresses (Ji et al., 2019; Wang et al., 2018). A higher ratio of positive-to-total interactions, but lower modularity, was identified in the subsurface snow network (Table 2). In general, higher positive interactions indicate increased microbial cooperation (Ju et al., 2014; Scheffer et al., 2012), whereas reduction in modularity indicates microbial niche-homogenization (Ji et al., 2019). The enhanced biotic associations and cooperation in the subsurface layer may be attributed to the occurrence of denitrification processes, as denitrification is a multi-step process that involves multiple bacterial cohorts to complete the process (Henry et al., 2005; Madsen, 2011; Yuan et al., 2021). The enhanced collaboration and deterministic succession was previously reported in bacterial community associated with the anoxic decomposition of microcystis biomass (Wu et al., 2020), while cross-feeding was shown to enhance positive interactions among the different members of the community (Borchert et al., 2021).

The path lengths of the subsurface network were lower than that of the surface layer (Table 2). The shorter path length has been proposed to be associated with a higher transfer efficiency of information and materials across the microorganisms in the network (Du et al., 2020), which are required for complex biological processes that require extensive bacterial collaboration, such as denitrification (Yuan et al., 2021). Thus, the short path length is consistent with the dominance of denitrification processes in the subsurface layer. Previous studies have proposed microbial interactions as biotic drivers that impact microbial diversity (Calcagno et al., 2017; Hunt and Ward, 2015). Thus, those microorganisms who are not adapted to the subsurface environment would be excluded from the environment, which provides an alternative explanation for the reduction in diversity (Scheffer et al., 2012; Ziegler et al., 2018; Bergk Pinto et al., 2019).

## 5 Conclusion

Our results showed the dynamics of nitrogen and bacterial community in supraglacial snow over nine days. Inorganic nitrogen was unchanged or slightly increased in the surface snow, while it decreased in subsurface snow. Due to atmospheric nitrogen deposition and potentially bacterial nitrogen fixation activities, nitrogen limitation is unlikely to occur in the surface snow. In contrast, nitrogen consumption was inferred in the subsurface snow. Nitrogen is traditionally recognized to be released from the supraglacial environment due to photolysis, whereas this study hints that nitrogen assimilation and denitrification could be alternative routes. Therefore, the increased nitrogen deposition due to anthropogenic activities may enhance the nitrogen consumption in the subsurface snow, which reduces the impact of increased nitrogen discharge on downstream glacier-fed rivers. In summary, our results provide a new perspective of the nutrients and bacterial community dynamics in supraglacial snow of the Tibetan Plateau. Further studies based on metagenome and metatranscriptome can enhance the understanding of bacterial functions.

*Data availability*. Sequence data generated in the present study have been deposited to the National Center for Biotechnology Information (NCBI) Sequence Read Archive under the ID PRJNA649151.

*Author contributions.* YL and MJ conceived the study and developed the idea. YC performed DNA extraction. YC and FW performed the environmental characterization measure. YC conducted the data statistical analysis. YC and KS wrote the first draft of the paper, and MJ, TV, and YL revised the paper substantially. All authors read and approved the final paper.

*Competing interests.* The authors declare that they have no conflict of interest.

*Acknowledgements.* We thank Zhengquan Gu, Paudel Adhikari Namita and Zhihao Zhang for their valuable input related to writing or providing maps of the sampling sites. We greatly thank Alexandre Magno Barbosa Anesio for revising our manuscript.

*Financial support.* This work was supported by the National Natural Science Foundation of China (grant number 91851207), the Second Tibetan Plateau Scientific Expedition and Research (STEP) program (grant number 2019QZKK0503), the National Key Research and Development Program of China (grant number 2019YFC1509103), and the Strategic Priority Research Program (A) of the Chinese Academy of Sciences (grant number XDA20050101).

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

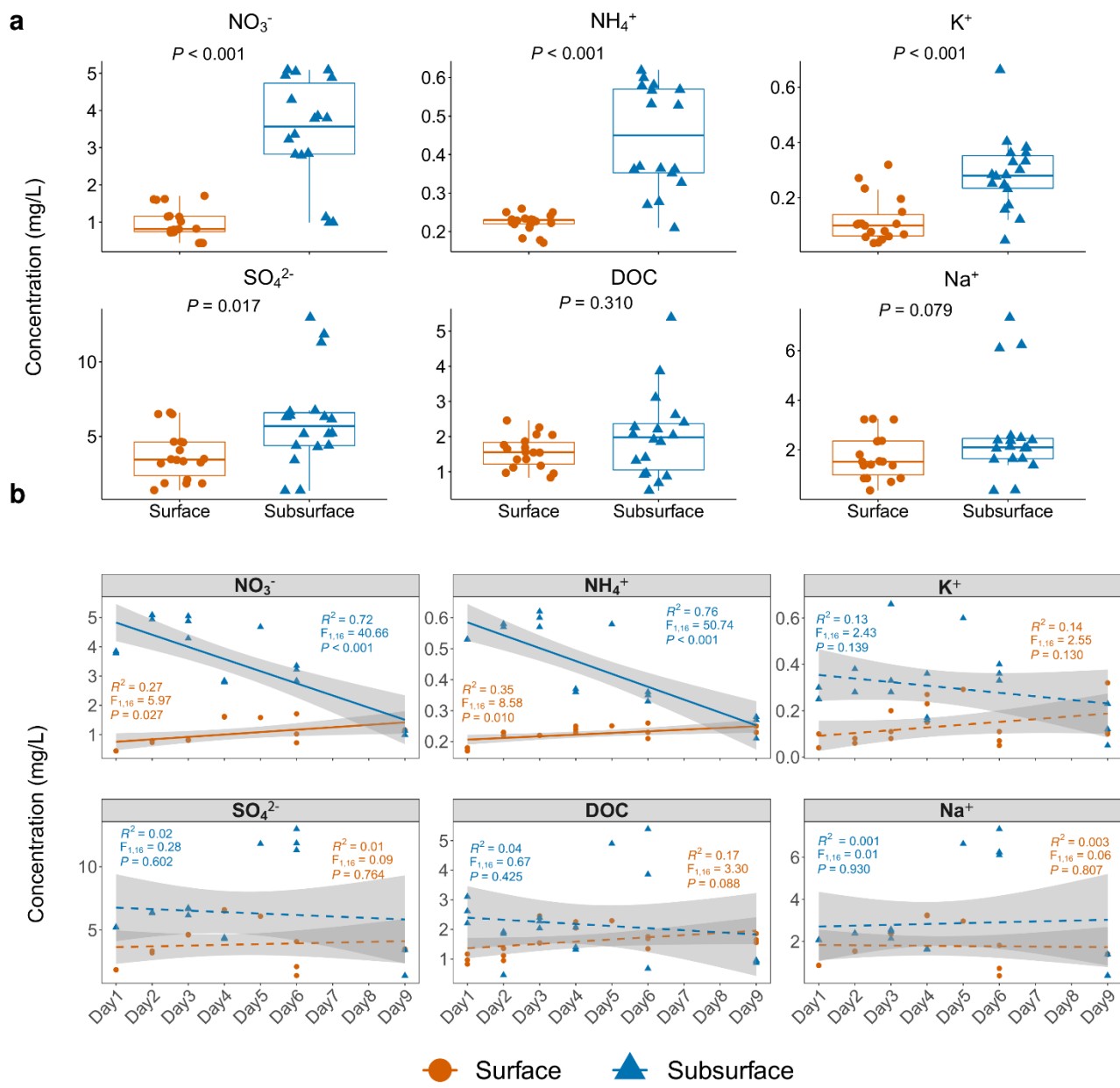

**Fig. 1 The pattern of environmental factors changes in the surface and subsurface snow layers.**
(a) Environmental factor comparisons in the surface and subsurface snow layers. Each dot represents an individual sample. Significantly higher concentrations of $NO_3^-$, $NH_4^+$, $K^+$, and $SO_4^{2-}$ were observed in the subsurface layer based on Wilcoxon rank-sum test. (b) Temporal changes of environmental factors in the surface and subsurface layers. The solid and dashed lines indicate significant and non-significant temporal changes, respectively. The concentration of $NO_3^-$ and $NH_4^+$ in the surface layer significantly increased with time while the concentration of $NO_3^-$, and $NH_4^+$, in the subsurface layer, significantly decreased with time. Significance is based on linear regression. Grey shading indicates the 95% confidence interval of regression.



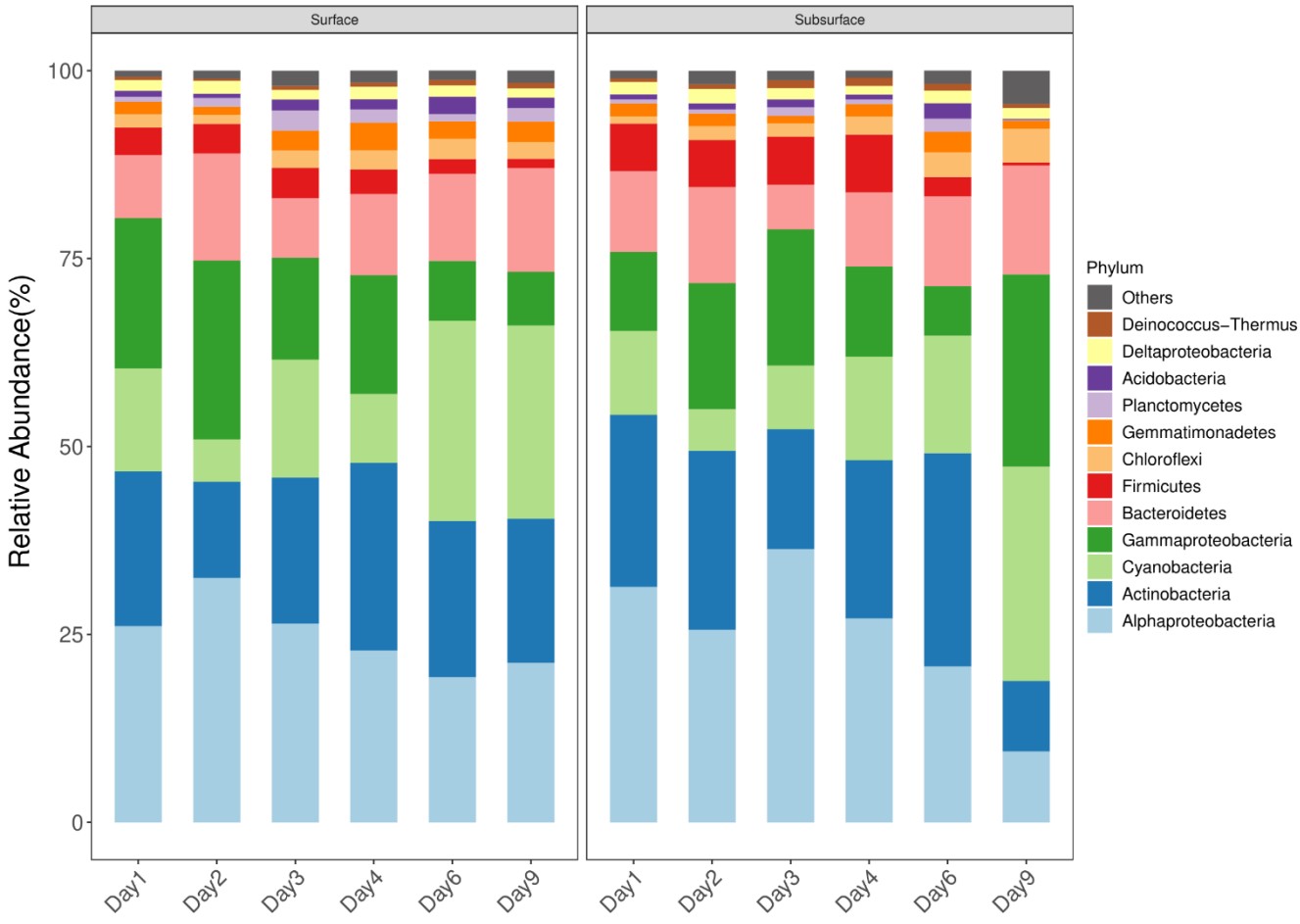

**Fig. 2 Taxonomic composition of bacterial community in snow.** Only dominant phyla are presented (relative abundance > 1%). The snow community are dominated by Alphaproteobacteria, Actinobacteria, Cyanobacteria, Gammaproteobacteria, Bacteroidetes, Firmicutes, Chloroflexi, Gemmatimonadetes, Planctomycetes, Acidobacteria, Deltaproteobacteria, and Deinococcus-Thermus.

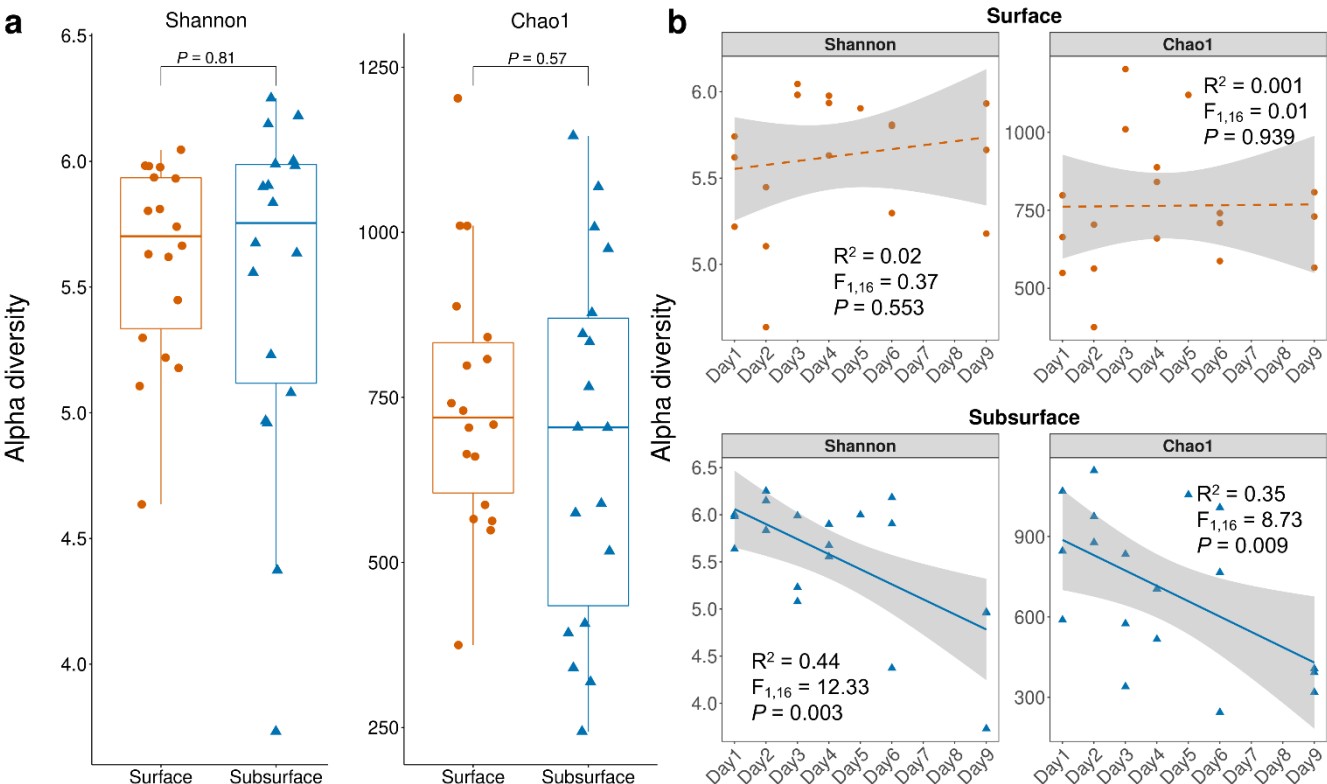

**Fig. 3 Bacterial alpha diversity in snow layers.** (a) Bacterial alpha diversity comparison between the surface and subsurface layers. Each dot represents an individual sample. For both Shannon and Chao1 indices, no significant difference was observed between the surface and subsurface snow layers. Comparison is based on Wilcoxon rank-sum test. (b) Temporal changes of the alpha diversity indices in the surface and subsurface snow layers. For the surface layer, no significant correlation was observed, while both Shannon and Chao1 showed a significantly reduction with time in the subsurface layer. Significance is based on linear regression. Grey shading indicates the 95% confidence interval of regression.

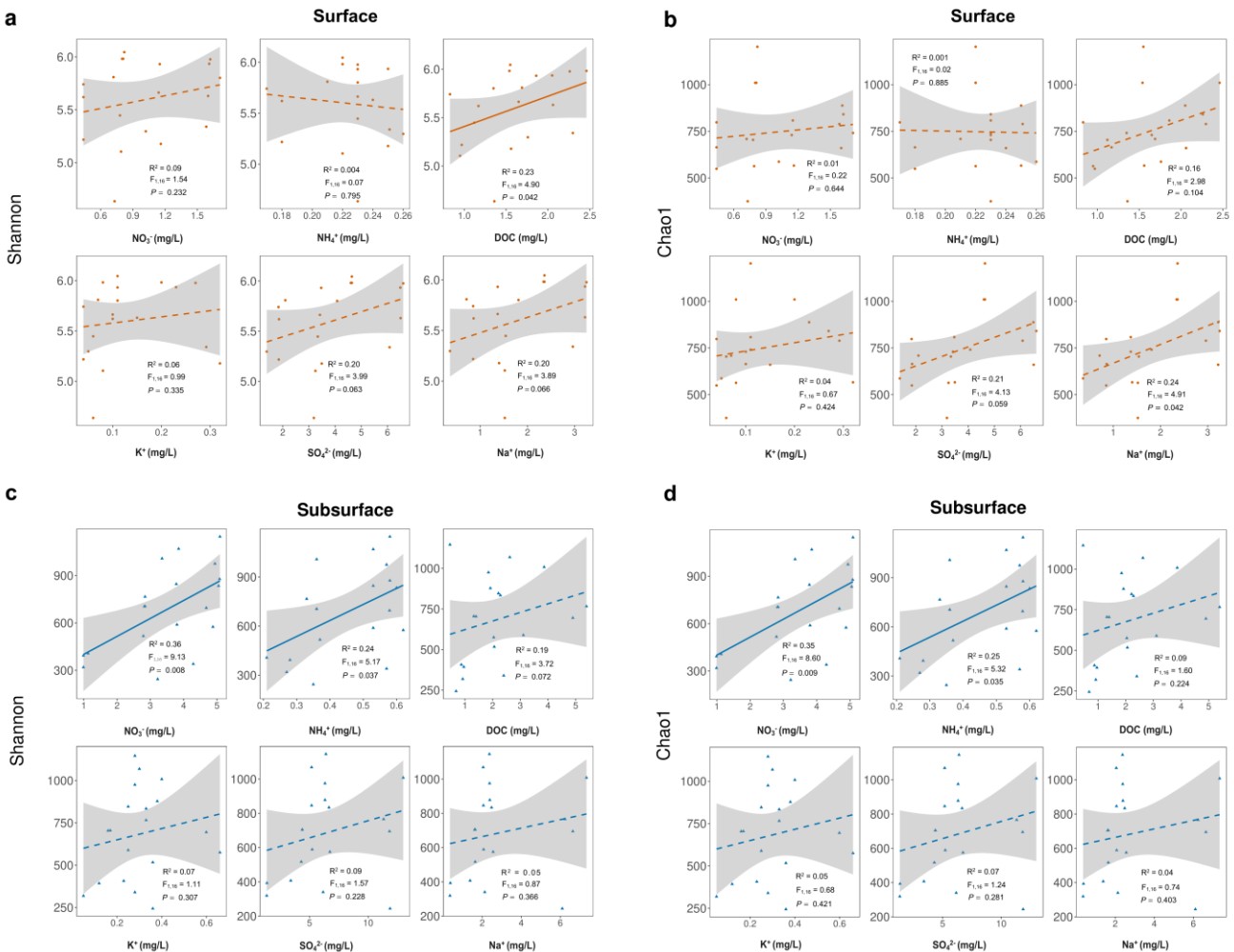

**Fig. 4 The influence of environmental factors on bacterial diversity.** Correlations of Shannon (a, c) and Chao1 (b, d) diversity indices with environmental factors in the surface and subsurface layers. Each dot represents an individual sample. The solid and dashed lines indicate significant and nonsignificant changes respectively. Significance is based on linear regression. Grey shading indicates the 95% confidence interval of regression.

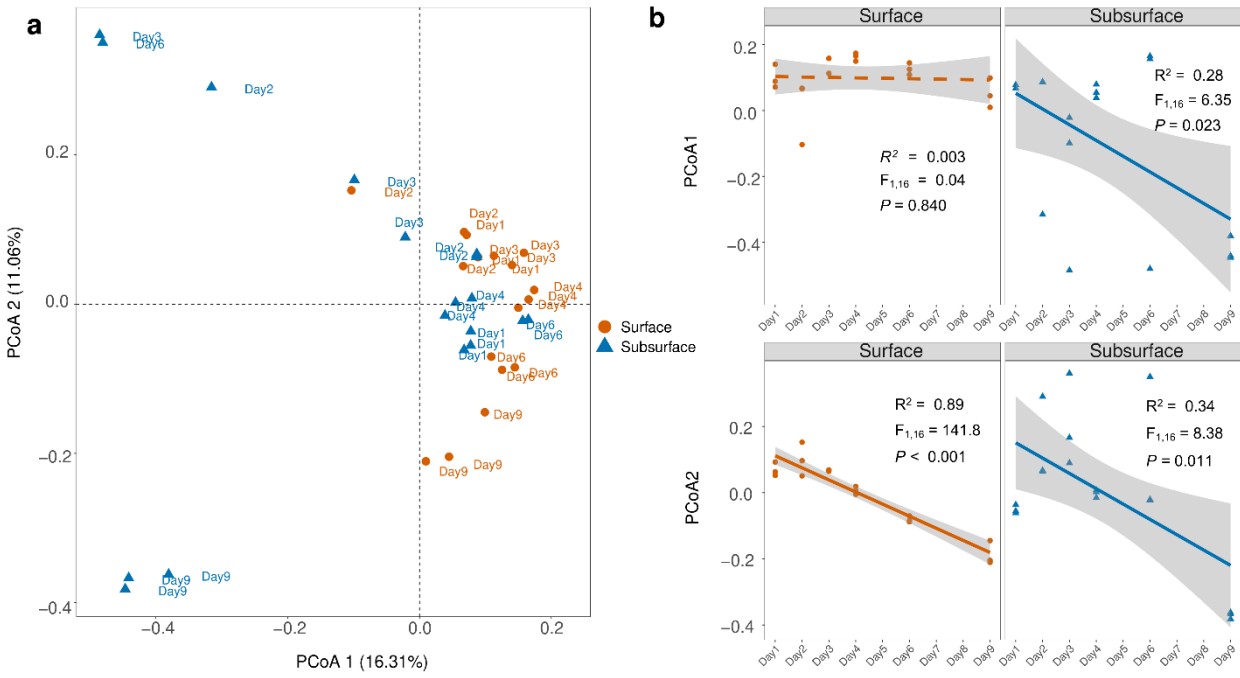

**Fig. 5 Principal coordinate analysis (PCoA) of microbial communities in the surface and subsurface snow.** (a) Bray-Curtis distance-based PCoA ordination plot. The microbial community structures of the surface and subsurface snows are significantly different (PERMANOVA, $P < 0.001$). (b) Pairwise regression analysis between PCoA scores and sampling time. The solid and dashed lines indicate significant and insignificant changes (based on linear regression), respectively. The PCoA1 scores for the bacterial community in the surface layer exhibit no significant correlation with time, while the PCoA2 scores significantly correlated with time. The PCoA1 and PCoA2 are both significantly correlated with time in the subsurface layer. Grey shading indicates the 95% confidence interval of regression.



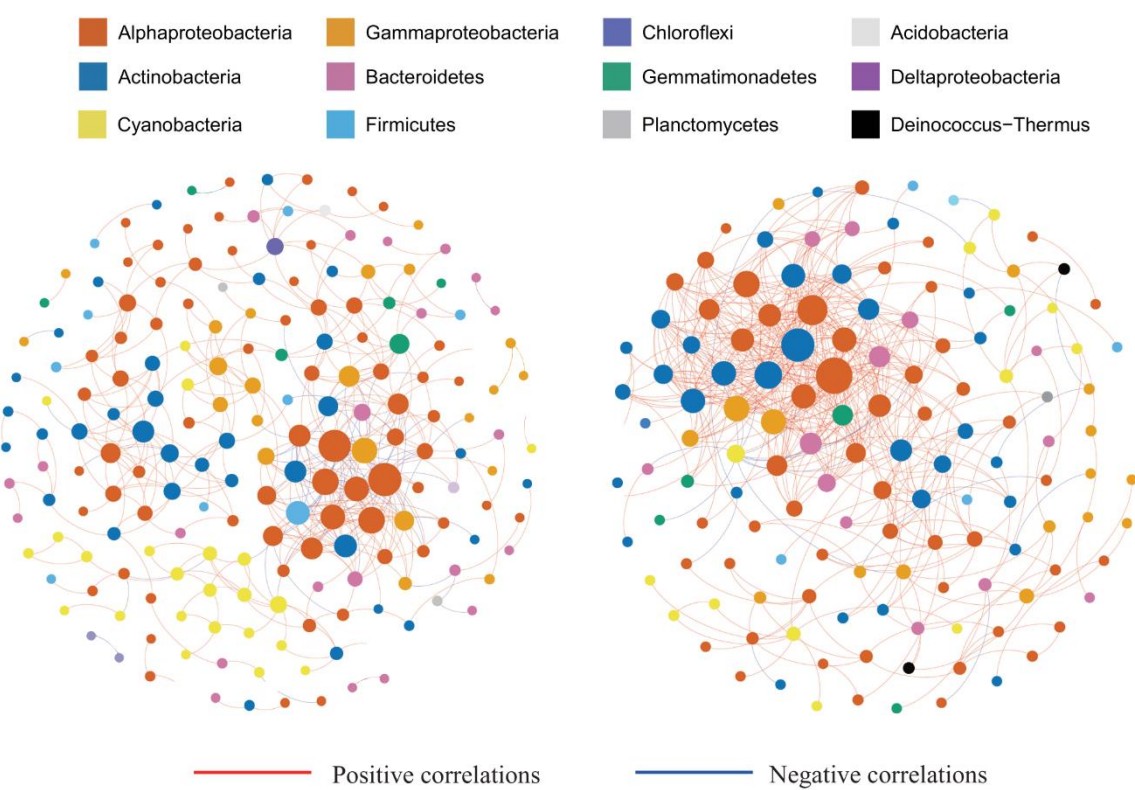


**Fig. 6 Bacterial Co-occurrence networks for the surface and subsurface layers communities.** Each node represents a bacterial amplicon sequence variant (ASV). The red solid lines represent positive correlations, and the blue solid lines represent negative correlations. Nodes are colored by taxonomy at the phylum level. The subsurface community networks are more complex with a higher positive-to-total correlation ratio.




**Table 1. Results of Mantel test showing the relationships between bacterial community composition and environmental factors in the surface and subsurface snow.** Significant correlations are in bold.

| Environmental factor | Surface | | Subsurface | |
|---|---|---|---|---|
| | R | *P* | R | *P* |
| $NO_3^-$ | 0.09 | 0.21 | 0.38 | **0.005** |
| $NH_4^+$ | 0.01 | 0.36 | 0.25 | **0.01** |
| DOC | 0.08 | 0.22 | 0.02 | 0.49 |
| $Na^+$ | 0.02 | 0.40 | 0.16 | 0.14 |
| $SO_4^{2-}$ | 0.00 | 0.44 | 0.25 | 0.09 |
| $K^+$ | 0.00 | 0.56 | 0.11 | 0.24 |


**Table 2. Topological properties of the empirical networks for the surface and subsurface bacterial communities.**

|  | Surface | Subsurface |
|---|---|---|
| No. of node | 197 | 140 |
| No. of edges | 436 | 523 |
| Number of edges per node | 2.21 | 3.73 |
| Positive links | 363 | 500 |
| Negative links | 73 | 22 |
| Ratio of positive-to-total interactions | 83% | 95% |
| Modularity | 0.65 | 0.40 |
| No. of modules | 23 | 12 |
| Average connectivity | 4.41 | 7.36 |
| Average clustering coefficient (avgCC) | 0.31 | 0.39 |
| Average path distance (GD) | 5.51 | 4.72 |
| Average degree (avgK) | 4.43 | 7.57 |
| Graph density | 0.02 | 0.06 |
| Transitivity (Trans) | 0.45 | 0.49 |
| Connectedness (Con) | 0.71 | 0.86 |