# Peer review of "Temporal variation of bacterial community and nutrients in Tibetan glacier snowpack"

_The Cryosphere, 2021_

## Author Comment (AC1)

**We thank Anonymous Referee #1 for their considered, detailed, and helpful review of our manuscript. Below we present our point-by-point responses.**

**Question 1.** Line 17 – word 'sub' missing I believe. Same sentence starting 'Our results' mixes past and present tense.

**Response:**

We apologize for the mistake. The spelling and the sentence have been changed to past tense.

Original manuscript:

Our results revealed dynamic bacterial communities in both surface and surface snow, and nitrogen is the key determinant of bacterial diversity, composition, community structure, and biotic interactions.

Amended manuscript (Lines 19-21):

Our results revealed rapid temporal changes in nitrogen (including nitrate and ammonium) and bacterial communities in both surface and subsurface snow.

**Question 2.** Line 18 – sentence starting 'Nitrate and ammonium'…this first past of this sentence is pretty vague – can you improve this with some data values please (or at least proportional change information).

**Response:**

We thank for this comment. The actual numbers have been added.

Original manuscript:

Nitrate and ammonium concentration increased and decreased in the surface and subsurface snow over time, therefore indicating accumulation and consumption processes, respectively.

Amended manuscript (Lines 21-24):

Nitrate and ammonium concentrations increased from 0.44 to 1.15 mg/L and 0.18 to 0.24 mg/L in the surface snow and decreased from 3.81 to 1.04 mg/L and 0.53 to 0.25 mg/L in the subsurface snow over time, therefore indicating accumulation and

consumption processes, respectively.

**Question 3.** Line 21 – sentence starting 'The nitrogen limitation'… place into past tense and remove 'the' throughout for clarity… e.g. 'Nitrogen limitation and dominance of denitrification in subsurface snow suggested stronger environmental and biotic filtering processes than in surface snow.'

**Response:**

We thank for this comment. The grammar mistake has been corrected as suggested.

Original manuscript:

The nitrogen limitation and the apparent dominance of the denitrification in the subsurface snow suggest stronger environmental and biotic filtering than those in the surface snow.

Amended manuscript (Lines 27-31):

The nitrogen limitation and enriched denitrification-related genes in subsurface snow suggested stronger environmental and biotic filtering than those in surface snow, which may explain the lower bacterial diversity, more pronounced community temporal changes, and stronger biotic interactions.

**Question 4.** Line 24: Sentence starting 'Collectively' – place into present tense and change 'revealed' to 'provides insight into nitrogen metabolism…'

**Response:**

We thank for this comment. The grammar mistakes have been corrected as suggested.

Original manuscript:

Collectively, these findings significantly advanced our understanding of microbial community variations and bacterial interactions after snow deposition, and revealed the dynamics of nitrogen metabolism in Tibetan snow.

Amended manuscript (Lines 30-32):

Collectively, these findings advance our understanding of bacterial community variations and bacterial interactions after snow deposition and provide a possible biological explanation for nitrogen dynamics in snow.

**Introduction**

**Question 5.** Lines 28 – 38: There is a mix of snow and glacier related information up front here (lines 28 – 38) – ie. Glaciers melting, and glacier ice being oligotrophic, but then snow related bacterial communities. I think it important to make clear the environment you are working with (presumably supraglacial snowpacks) and try to not mix information between snow and glacier ice itself, which also houses diverse autotrophic and heterotrophic populations that undertake significant biogeochemical cycling of carbon and major nutrients.

**Response:**

We thank for the comment, and we have rewritten the first paragraph of the introduction and the manuscript now focuses only on the supraglacial snow to ensure the information is clear to the readers.

Amended manuscript (Lines 35-47):

Global warming accelerates glacier melting across the globe, supraglacial snow is particularly vulnerable (Hodson et al., 2008) with the carbon and nitrogen stored being released into downstream ecosystems in meltwaters (Wadham et al., 2019; Hodson et al., 2005). The composition and abundance of nutrients in supraglacial snows are regulated by glacier-dwelling microorganisms (Hodson et al., 2008). A range of metabolically active bacteria have been reported in supraglacial snow, including Bacteroidetes, Actinobacteria, Firmicutes, and Alphaproteobacteria (Miteva, 2008; Maccario et al., 2019; Carey et al., 2016; Lazzaro et al., 2015; Michaud et al., 2014). These microorganisms perform key ecological functions in biogeochemical cycling such as carbon and nitrogen fixation, which are vital to the nutrient-limited supraglacial ecosystem. Changes in their community composition and activities will influence the dynamics of nutrient storage, transformation, and release. Thus, it is crucial to understand how the bacterial community in supraglacial snow changes across time and to determine whether those changes are associated with the temporal nutrient dynamics in snow.

**Question 6.** Lines 39 - 46: again, I think you need to be more explicit here that you are presumably talking about precipitation in the ablation zone of glaciers….i.e. snow fall that ablates in the subsequent melt season as opposed to that which is accumulated and eventually turned into glacial ice.

**Response:**

We thank for the comment. In our study, we attempted to reveal the dynamics of nutrients and bacterial community in supraglacial snow of fresh snow across a short temporal scale. Previous studies have investigated nutrients and microbial community structure changes across the entire ablation period or seasonal changes, such as Larose et al. (2013)[1] revealed that the form of nitrogen varied as a function of time in supraglacial snow during a two-month field study at the Svalbard, Norway. However, several snowfall events would be expected in those studies. Thus, the observations could be an accumulation effect, whereas focusing on fresh snow after a single snowfall is still yet elusive. We apologize for not stating the aim of our study clear enough, and we have amended the introduction to emphasize the aim of our study.

1. Larose, C., Dommergue, A., and Vogel, T. M.: Microbial nitrogen cycling in Arctic snowpacks, Environmental Research Letters, 8, 035004, 10.1088/1748-9326/8/3/035004, 2013

Amended manuscript (Lines 50-63):

Several studies investigated the dynamics of nutrient and bacterial changes in supraglacial snow during the ablation period. Larose et al. (2013a) revealed that the form of nitrogen varied as a function of time in supraglacial snow during a two-month field study at the Svalbard, Norway and fluctuations in microbial community structure have been reported with the relative abundance of fungi and bacteria (such as Bacteroidetes and Proteobacteria) increased and decreased, relatively. Seasonal shifts in snowpack bacterial communities have also been reported in the mountain snow in Japan, rapid microbial growth was observed with increasing snow temperature and meltwater content (Segawa et al., 2005). However, the results of these studies are likely the consequence of these several precipitation events due to the long period of time. During precipitation, a new snow layer forms above the previous ones, which is

responsible for the stratified snowpack structure. These different snow layers have distinct physical and chemical characteristics and their age also differed substantially (Lazzaro et al., 2015). Thus, the microbial process across the aged snowpack could be complex, whereas focusing on supraglacial snow from a single snowfall event could provide unique insights into the bacterial and nutrient dynamics. Hell et al. (2013) reported bacterial community structure changes during the ablation period across five days in the high Arctic, while the bacterial and nutrient dynamics during the snow accumulation period remain elusive.

**Question 7.** The description of dominant groups is limited here – Cyanobacteria were shown by Carey et al. 2016 to dominate the surface (0-15cm) of a sub-Alpine snowpack based on 16S sequencing, but it is widely known that numerous eukaryotic microalgae can dominate surface snow on and off glaciers. Please expand this information to include the suite of microbial communities that are known to dominate snowpacks or alternatively change the wording throughout to focus only on bacterial communities as opposed to 'microbial communities.

**Response:**

We appreciate the reviewer for raising this concern. This paper only focuses on bacterial communities, so we have changed the word to "bacterial communities" instead of "microbial communities".

Amended manuscript (Lines 64-70):

Surface and subsurface typically have distinct bacterial community structures due to the environmental filtering from the vertical profile of temperature, solar radiation intensity, and nutrients (Xiang et al., 2009; Møller et al., 2013; Carey et al., 2016). For example, Cyanobacteria tend to dominate upper snow layers (0-15 cm) (Carey et al., 2016), while their relative abundance is greatly reduced in the deeper snow layer (Xiang et al., 2009). This is likely due to the lower light intensity in the deeper snow, which favors heterotrophic bacteria such as the Actinobacteria and Firmicutes. Differences in physicochemical conditions can also indirectly influence bacterial community structure through impacts on the types of biotic interactions that dominate an environment (Friedman and Gore, 2017; Khan et al., 2018; Bergk Pinto et al., 2019). For example,

the addition of organic carbon shifted bacterial interactions from collaboration to competition in Arctic snow (Bergk Pinto et al., 2019). In comparison, intensive collaboration can enhance complex organic carbon degradation and mineralization, which are particularly important for oligotrophic environments such as glaciers (Krug et al., 2020). Collaboration is also known to be essential to biological processes such as ammonia oxidation and denitrification, in which various organisms carry out different steps of these processes (Henry et al., 2005; Madsen, 2011; Yuan et al., 2021). These changes in interactions and network complexity can favor or disadvantage certain bacterial groups, thereby changing the bacterial community structure (i.e., biofiltering).

**Question 8.** I find the sentences at the end of this paragraph contradicting. You state that differences in physiochemical conditions shape community structure (with the previous example of cyanos in surface and not in subsurface snow), but then state whether microbes in different snowpack layers show similar responses to environmental selection is still largely unknown. Can you clarify this please?

**Response:**

Thank you for your comment The sentence at the end of the paragraph intended to propose that whether the different microbial communities will have distinct microbial functions is largely unknown. We have amended the sentence to correct this.

Amended manuscript (Lines 59-61):

Thus, the microbial process across the aged snowpack could be complex, whereas focusing on supraglacial snow from a single snowfall event could provide unique insights into the bacterial and nutrient dynamics.

**Question 9.** Lines 56 – 60: add "The" before 'Tibetan Plateau' in the first sentence please.

**Response:**

We thank for this comment, the grammar mistake has been corrected as indicated.

Original manuscript:

Tibetan Plateau is the world's third-largest ice reservoir, after those in Antarctica and Greenland (Qiu, 2012).

**Question 10.** Line 58 sentence starting 'The glacier melting' – this is a very vague statement with no references…can you please add details and appropriate references to evidence this.

**Response:**

We thank for this comment. We have added details and appropriate references to evidence this.

Original manuscript:

Tibetan Plateau is the world's third-largest ice reservoir, after those in Antarctica and Greenland (Qiu, 2012). It is warming at a rate twice of the global average (Chen et al., 2015), causing rapid shrinkage of glaciers and snow (Rauscher et al., 2007; Hall and Fagre, 2003). The glacier melting leads to the enhanced discharge of microorganisms and nutrients into downstream aquatic and terrestrial ecosystems, which makes an impact on their biogeochemical processes.

Amended manuscript (Lines 81-86):

The Tibetan Plateau is the world's third-largest ice reservoir, after those in Antarctica and Greenland (Qiu, 2012). It is warming at a rate twice the global average (Chen et al., 2015), and 95% of the Tibetan glaciers retreated between 1990 to 2005 (Rauscher et al., 2007; Hall and Fagre, 2003; Yao et al., 2007). Glacier melting increases the discharge of microorganisms and nutrients in meltwater into downstream aquatic ecosystems (Kohler et al., 2020), which substantially impacts the bacterial community and biogeochemical processes (Liu et al., 2021).

**Question 11.** I would recommend to blend this paragraph with the subsequent paragraph and add in references to sentences where needed (e.g. ablation zone microbes having a greater impact on downstream processes).

**Response:**

We thank the reviewer for this constructive comment. We have blended this paragraph with the subsequent paragraph and have rewritten our research purpose as follows.

Amended manuscript (Lines 81-99):

The Tibetan Plateau is the world's third-largest ice reservoir, after those in Antarctica and Greenland (Qiu, 2012). It is warming at a rate twice the global average (Chen et al., 2015), and 95% of the Tibetan glaciers retreated between 1990 to 2005 (Rauscher et al., 2007; Hall and Fagre, 2003; Yao et al., 2007). Glacier melting increases the discharge of microorganisms and nutrients in meltwater into downstream aquatic ecosystems (Kohler et al., 2020), which substantially impacts the bacterial community and biogeochemical processes (Liu et al., 2021). Thus, it is crucial to understand the transformation processes of the bacterial community and nutrients in the supraglacial snow. Several studies have investigated the nutrient and bacterial community changes in supraglacial snow across the winter (Brooks et al., 1998; Liu et al., 2006), but the bacterial and nutrient dynamics of freshly fallen snow have been largely overlooked. These short temporal changes will influence the following post-depositional processes after it is buried by the next snowfall, and will ultimately determine the physicochemical properties of the stratified snow in the following year. In the present study, we investigated the bacterial community and snow physiochemical property changes in the surface and subsurface supraglacial snow during a nine-day period after a single snowfall event at the Dunde Glacier on the northeast of the Tibetan Plateau. We aimed to answer the following key questions: 1) do the bacterial community and nutrient changes in a short temporal scale, 2) do the bacterial communities in different snow layers exhibit similar community temporal changes, and 3) are the temporal changes in the surface and subsurface snow related to environmental filtering, biotic interactions, or both?

**Question 12.** The research questions seem sound but you have a strong emphasis on temporal changes, and are only monitoring here for 9 days – arguably a very small time slice of a snowpack's melt period. How will this limited data be representative of larger temporal patterns?

**Response:**

We appreciate the reviewer for raising this concern. We agree that a nine days observation cannot reflect the nutrient changes for the entire snowmelt period. We have narrowed the scope of the present study, and focus only on the bacterial and nutrient changes from a single snowfall event across a short temporal scale, which has rarely been investigated. We have added a paragraph to introduce the knowledge gap in bacterial and nutrient dynamics in supraglacial snow over a short temporal scale, and rephrase the scientific questions to reflect this.

Amended manuscript (Lines 50-63):

Several studies investigated the dynamics of nutrient and bacterial changes in supraglacial snow during the ablation period. Larose et al. (2013a) revealed that the form of nitrogen varied as a function of time in supraglacial snow during a two-month field study at the Svalbard, Norway and fluctuations in microbial community structure have been reported with the relative abundance of fungi and bacteria (such as Bacteroidetes and Proteobacteria) increased and decreased, relatively. Seasonal shifts in snowpack bacterial communities have also been reported in the mountain snow in Japan, rapid microbial growth was observed with increasing snow temperature and meltwater content (Segawa et al., 2005). However, the results of these studies are likely the consequence of these several precipitation events due to the long period of time. During precipitation, a new snow layer forms above the previous ones, which is responsible for the stratified snowpack structure. These different snow layers have distinct physical and chemical characteristics and their age also differed substantially (Lazzaro et al., 2015). Thus, the microbial process across the aged snowpack could be complex, whereas focusing on supraglacial snow from a single snowfall event could provide unique insights into the bacterial and nutrient dynamics. Hell et al. (2013) reported bacterial community structure changes during the ablation period across five days in the high Arctic, while the bacterial and nutrient dynamics during the snow accumulation period remain elusive.

And (Lines 83-99)

Glacier melting increases the discharge of microorganisms and nutrients in meltwater into downstream aquatic ecosystems (Kohler et al., 2020), which substantially impacts the bacterial community and biogeochemical processes (Liu et al., 2021). Thus, it is crucial to understand the transformation processes of the bacterial community and nutrients in the supraglacial snow. Several studies have investigated the nutrient and bacterial community changes in supraglacial snow across the winter (Brooks et al., 1998; Liu et al., 2006), but the bacterial and nutrient dynamics of freshly fallen snow have been largely overlooked. These short temporal changes will influence the following post-depositional processes after it is buried by the next snowfall, and will ultimately determine the physicochemical properties of the stratified snow in the following year. In the present study, we investigated the bacterial community and snow physiochemical property changes in the surface and subsurface supraglacial snow during a nine-day period after a single snowfall event at the Dunde Glacier on the northeast of the Tibetan Plateau. We aimed to answer the following key questions: 1) do the bacterial community and nutrient changes in a short temporal scale, 2) do the bacterial communities in different snow layers exhibit similar community temporal changes, and 3) are the temporal changes in the surface and subsurface snow related to environmental filtering, biotic interactions, or both?

**Methods**

**Question 13**. Line 71 – remove 'the' before October.

**Response:**

The grammar mistake has been corrected.

Original manuscript:

Snow samples were collected from the ablation zone at Dunde glacier (38°06′N, 96°24′E, 5325 m above the sea-level), during the October and November, 2016 (Supplementary Fig S1).

Amended manuscript (Lines 102-103):

Snow samples were collected from the ablation zone at Dunde glacier (38°06′N, 96°24′E, 5325 m above the sea-level), during October and November, 2016

(Supplementary Fig. S1).

**Question 14.** Line 74 – OK, so actually not daily sampling, but spread over nine days. Also, your dates and day numbers don't match – you have 6 dates that you list, and 7 'days' that you list….

**Response:**

We apologize for the misunderstanding. There was a typo in the original sentence, and this has been corrected.

Original manuscript:

Sampling was conducted over a nine-day period (on the 24th, 25th, 26th, 27th, and 29th of October, and the 2nd November, which are referred as day 1, 2, 3, 4, 5, 6, and 9 thereafter).

Amended manuscript (Lines 107-110):

Snow samples were collected on the 24th, 25th, 26th, 27th, and 29th of October, and the 2nd November (which are referred as day 1, 2, 3, 4, 6, and 9) until the next snowfall started.

**Question 15.** Line 75: sampling during October and November presumably means recently deposited snow, i.e. snow from the start of the autumn/winter season. Thus you are sampling relatively fresh snow, rather than e.g. snow just before the start of the next melt season, ie. the snow that would be released into the (downstream) environment. How are these data then reflective of the type of microbial communities and biogeochemistry that are important for downstream ecosystems as asserted earlier in the manuscript? Will the communities and nutrient load not continue to change significantly throughout the whole autumn/winter/spring season until the onset of melt? This links back to my previous question on whether a 9-day sampling period is big enough to address the larger questions being related to here.

**Response:**

We appreciate the reviewer for raising this comment. We agree that changes in this nine-day can't directly impact the nutrient released during the ablation period of the

following year, thus it is insufficient to address such a big question. We have altered the goal of the present study, which now attempts the fill a missing but important knowledge gap in nutrient and microbial community changes immediately after a snowfall. We believe this information is utterly needed as this freshly fallen snow will be buried during the next snowfall, which will be responsible for the stratified snowpack structure, and will collectively influence the nutrient release during the coming ablation period. We have amended the introduction to emphasize the value and importance of carrying out a short temporal scale investigation on bacterial community and nutrients.

Amended manuscript (Lines 50-63):

Several studies investigated the dynamics of nutrient and bacterial changes in supraglacial snow during the ablation period. Larose et al. (2013a) revealed that the form of nitrogen varied as a function of time in supraglacial snow during a two-month field study at the Svalbard, Norway and fluctuations in microbial community structure have been reported with the relative abundance of fungi and bacteria (such as Bacteroidetes and Proteobacteria) increased and decreased, relatively. Seasonal shifts in snowpack bacterial communities have also been reported in the mountain snow in Japan, rapid microbial growth was observed with increasing snow temperature and meltwater content (Segawa et al., 2005). However, the results of these studies are likely the consequence of these several precipitation events due to the long period of time. During precipitation, a new snow layer forms above the previous ones, which is responsible for the stratified snowpack structure. These different snow layers have distinct physical and chemical characteristics and their age also differed substantially (Lazzaro et al., 2015). Thus, the microbial process across the aged snowpack could be complex, whereas focusing on supraglacial snow from a single snowfall event could provide unique insights into the bacterial and nutrient dynamics. Hell et al. (2013) reported bacterial community structure changes during the ablation period across five days in the high Arctic, while the bacterial and nutrient dynamics during the snow accumulation period remain elusive.

Glacier melting increases the discharge of microorganisms and nutrients in meltwater into downstream aquatic ecosystems (Kohler et al., 2020), which substantially impacts the bacterial community and biogeochemical processes (Liu et al., 2021). Thus, it is crucial to understand the transformation processes of the bacterial community and nutrients in the supraglacial snow. Several studies have investigated the nutrient and bacterial community changes in supraglacial snow across the winter (Brooks et al., 1998; Liu et al., 2006), but the bacterial and nutrient dynamics of freshly fallen snow have been largely overlooked. These short temporal changes will influence the following post-depositional processes after it is buried by the next snowfall, and will ultimately determine the physicochemical properties of the stratified snow in the following year. In the present study, we investigated the bacterial community and snow physiochemical property changes in the surface and subsurface supraglacial snow during a nine-day period after a single snowfall event at the Dunde Glacier on the northeast of the Tibetan Plateau. We aimed to answer the following key questions: 1) do the bacterial community and nutrient changes in a short temporal scale, 2) do the bacterial communities in different snow layers exhibit similar community temporal changes, and 3) are the temporal changes in the surface and subsurface snow related to environmental filtering, biotic interactions, or both?

**Question 16.** Line 80: You have matched the same vertical distribution profile of Carey et al. 2016 (i.e. upper 15 cm and 15 – 30cm) – can you justify why these layer depths were appropriate for a completely different snowpack with presumably different physiochemical conditions? i.e. do you have any density measurements etc to quantify your layer characteristics?. Snowpacks can have numerous different layer profiles dependent on precipitation and subsequent metamorphism processes.

**Response:**

We thank for this comment. Unfortunately, we didn't measure the density of the two snow layers. This snowpack was fresh snow from one snowfall event, and we didn't see a clear physical difference between the two layers at least within the nine days

period, thus if there were any metamorphism changes, they would have happened after our sampling started. Each snow pit was approximately 30 cm deep at the beginning of sampling. To get enough DNA for sequencing, we divided the snowpack equally into surface and subsurface layers (approximately 15 cm deep for each layer), according to Carey (2016). We amended the method section to explain the sampling design.

Amended manuscript (method section) (Lines 105-113):

No supraglacial snow was observed on the glacier surface on the 10[th] of October when first arrived at the camp. Snowfall started on the 18[th] and ended on the 23[rd] of October. Sampling was conducted over a nine-day period after the snowfall stopped on a flat 5 m × 3 m small area to reduce the impact of sample heterogeneity due to spatial variations. Snow samples were collected on the 24[th], 25[th], 26[th], 27[th], and 29[th] of October, and the 2[nd] November (which are referred as day 1, 2, 3, 4, 6, and 9) until the next snowfall started. This enabled us to follow the development of bacterial communities and the chemical environment through time after deposition. The ambient air temperature at the sampling period is averaged -8 °C (data available through the European Centre for Medium-Range Weather Forecasts, Supplementary Fig. S2), no snow melting was observed over the nine days.

**Question 17.** Line 82/83: just to clarify – a total of n = 3 surface and subsurface snow samples, and n = 3 surface and subsurface samples for aqueous geochemistry were taken on each sampling day?

**Response:**

This is correct, we collected three samples each for the surface and subsurface snow, and each sample was divided into two portions, 100 ml of snow was used for physiochemical analyses and the remaining 3L was used for DNA extraction.

Original manuscript:

For each snow pit, the top 1 cm in contact with the air was removed using a sterile spoon to avoid contamination, and then surface and subsurface snow were collected using a sterilized Teflon shovel into 3L separate, sterile sampling bags.

Amended manuscript (Lines 118-122):

For each snow pit, the top 1 cm in contact with the air was removed using a sterile spoon to avoid contamination, and then surface and subsurface snow were collected using a sterilized Teflon shovel into 3 L sterile sampling bags separately. Approximately 100 mL were used for physicochemical analyses, whereas the rest was used for DNA extraction.

**Question 18.** A nice level of detail throughout for DNA extraction, amplification, sequencing, sequence processing and identification and subsequent analyses.

**Response:**

Thank you for the comment.

**Results:**

**Question 19.** Section 3.1. Can you clarify the nitrogen data in figure 1 please? It appears that the total nitrogen (presumably inorganic + organic N fractions) is much lower than the corresponding nitrate concentration. It states in the methods that different methods were used for TN, NH4 and NO3 quantification. How do you account for this ~4-fold lower TN concentration than NO3 concentration for example? Unless the TN data reflect just the organic fraction after subtraction of inorganics? This isn't clear. Given the focus on N in the manuscript this needs to be clear.

**Response**:

We are extremely grateful for the reviewer to point this out. As the reviewer's comment below suggested, the range of the standard curve is much smaller than the total nitrogen we reported, and this is most likely to cause the underestimation of total nitrogen. Therefore, we have removed the results regarding total nitrogen throughout the manuscript, and this does not significantly impact our conclusion, i.e., nitrate and ammonium concentrations covaried with the microbial diversity and community structure in subsurface snow.

**Question 20.** Looking at Supp Fig. S2, the standard curve for TN goes up to ~0.4 mg/L but you are presenting data up to ~ 1.25 mg/L in the manuscript, well above this level.

Have you calculated TN concentration from peak areas much greater than the standard curve bounds you have performed? Also, please quantify the relationships on Figure S2 with linear regression, not correlation.

**Response:**

We appreciate the reviewer for raising this concern. We agree that the total nitrogen has been underestimated due to the incorrect standard curve range, and the results regarding the total nitrogen have been removed completely. We have performed the linear regression using the "lm" function in R, the results are presented in the supplementary figure.

Amended manuscript (Supplementary Fig S3):

[Figure]

Fig S3 Standard curve of $NO_3^-$, $NH_4^+$ ions. The x-axis is the concentration of the standard sample; the y-axis represents the peak area. Significance is based on linear regression.

**Question 21.** The 'significant increases' in NO3 and NH4 through time are rather small in magnitude – please include the actual R value in the text to illustrate the effect size of this relationship.

**Response:**

We thank for the comment. We have rewritten the sentence, added the R-value in the manuscript, and also performed linear regression analyses.

Original manuscript:

The concentration of $NO_3^-$, and $NH_4^+$ ions in the surface layer significantly increased with time (Fig 1b). In comparison, the concentrations of TN, $NO_3^-$, and $NH_4^+$ in the subsurface layer significantly decreased with time.

Amended manuscript (Lines 224-228):

The concentration of $NO_3^-$ and $NH_4^+$ ions in the surface snow increased with time ($F_{1,16}$ = 5.97, $P$ = 0.027, $R^2$ = 0.27 and $F_{1,16}$ = 8.58, $P$ = 0.010, $R^2$ = 0.35, respectively, Fig. 1b). In comparison, they decreased with time in the subsurface snow ($F_{1,16}$ = 40.66, $P$ < 0.001, $R^2$ = 0.72 and $F_{1,16}$ = 50.74, $P$ < 0.001, $R^2$ = 0.76, respectively). Other environmental factors exhibited no significant changes with time.

**Question 22.** How can you be certain the concentration of N has changed through time given that you are not repeat sampling in the same locations? The methods detail that new snow pits were dug each sampling day (as would be expected). How can you be certain that the variability 'through time' in your data, does not simply reflect variability in space, i.e. heterogeneity in snowpack N content across your sampling area?

**Response:**

We thank the reviewer for this comment. We believe that spatial heterogeneity is unavoidable in a field study. Nevertheless, the sampling strategy minimized the possibility that the pattern observed is solely due to the "variability in space". Firstly, the sampling area was on a relatively small flat area, which could minimize the impact of spatial variations. Secondly, the site where three snowpits were dug every day was randomly picked to avoid the pattern caused by spatial variation.

**Section 3.2.**

**Question 23.** Again, I find the presentation and reliance on correlations between diversity indices / relative abundance and time / Geochem data to be a bit misleading. These correlations are rather weak but the phraseology presents them as 'significant' changes. Can you reword presentation of correlations to reflect their actual strength, e.g. "a negative correlation was apparent between Shannon and Chao1 indices and time in the subsurface layer".

**Response:**

We appreciate the reviewer for raising this concern. We rephrased the sentences and performed additional linear regression analyses.

Original manuscript:

The bacterial Shannon and Chao1 indices in the surface layer were $5.61 \pm 0.39$ and $744 \pm 199$, respectively, which were not significantly different from those in the subsurface layer ($5.52 \pm 0.68$ and $705 \pm 269$, respectively) (Fig 2a). In the surface layer, the Shannon and Chao1 indices did not change significantly with time (Pearson correlation; r= 0.15, $P = 0.553$; r = -0.02, $P = 0.939$, respectively, Fig 2b). In contrast, both the Shannon and Chao1 indices both significantly decreased with time in the subsurface layer (Pearson correlation; r = 0.63, $P = 0.003$; r = 0.56, $P = 0.009$, respectively). In the surface layer, the Shannon indices positively correlated with the concentration of DOC (Pearson correlation; r= 0.48, $P = 0.04$, Fig 3a). In the subsurface layer, the Shannon and Chao1 indices were positively correlated with the concentration of TN, $NO_3^-$ and $NH_4^+$ (Pearson correlation; $P < 0.05$, Fig 3b and Supplementary Fig S4).

Amended manuscript (Lines 257-272):

The bacterial Shannon and Chao1 indices in the surface snow were $5.61 \pm 0.39$ and $744 \pm 199$, respectively, which were not significantly different from those in the subsurface layer ($5.52 \pm 0.68$ and $705 \pm 269$, respectively) ($P = 0.81$ and $0.57$, respectively) (Fig. 3a). In the surface snow, the Shannon and Chao1 indices were similar across the nine days ($F_{1,16} = 0.37$, $P = 0.553$, $R^2 = 0.02$ and $F_{1,16} = 0.01$, $P = 0.939$, $R^2 = 0.001$, respectively; Fig. 3b). In comparison, negative associations were observed in both Shannon and Chao1 indices with time in the subsurface snow ($F_{1,16} = 12.33$, $P = 0.003$, $R^2 = 0.44$ and $F_{1,16} = 8.73$, $P = 0.009$, $R^2 = 0.35$, respectively). In the surface layer, the positive correlations of Shannon and Chao1 indices with the DOC and sodium ions were apparent ($F_{1,16} = 4.90$, $P = 0.042$, $R^2 = 0.23$ and $F_{1,16} = 4.91$, $P = 0.042$, $R^2 = 0.24$, respectively; Fig. 4a,b). In the subsurface snow, the positive correlations of Shannon and Chao1 indices with the concentrations of $NO_3^-$ and $NH_4^+$ were apparent (Shannon diversity: $F_{1,16} = 9.13$, $P = 0.008$, $R^2 = 0.36$ and $F_{1,16} = 5.17$, $P = 0.037$, $R^2 = 0.24$, respectively; Chao1 index: $F_{1,16} = 8.60$, $P = 0.009$, $R^2 = 0.36$ and $F_{1,16} = 5.32$, $P = 0.035$,

$R^2 = 0.25$, respectively; Fig. 4cd). This is consistent with the random forest analysis results, which identified the concentrations of $NO_3^-$ and $NH_4^+$ as the significant determinants of bacterial Shannon diversity in the subsurface layer (Supplementary Fig. S8).

**Question 24.** Can multiple linear regressions be tested here to account for differences in diversity taking into account multiple factors simultaneously rather than simplistic single variable correlations?

**Response:**

We thank for the constructive suggestion. We have performed additional Random Forest analysis to investigate the influence of environmental factors, which takes the influence of multiple variables into account. This result is consistent with the correlation analysis results, which identified the concentration of nitrate and ammonium as the significant determinants of bacterial diversity in the subsurface snow layer. We added this result in the Result section and supplementary as Figure S9.

Amended manuscript (Lines 270-272):

This is consistent with the random forest analysis results, which identified the concentrations of NO3- and NH4+ as the significant determinants of bacterial Shannon diversity in the subsurface layer (Supplementary Fig. S8).

[Figure]

Percentage increase in Mean Square Error (MSE)

Fig S8 Random Forest analysis results showing the contribution of environmental factors on the bacterial Shannon and Chao1 indices in the surface and subsurface snow. (Red bars represent significant indicators at $P < 0.05$, grey bars represent non-significant indicators).

**Question 25.** Please move the second paragraph on the actual community composition to the start of this section, which should then be followed by comparisons to other datasets.

**Response:**

We thank for the suggestion, and we have moved the second paragraph to the start of this section. We added the comparison in the discussion part.

Amended manuscript (Lines 320-323):

The surface and subsurface snow were both dominated by Alphaproteobacteria, Actinobacteria, Cyanobacteria, Gammaproteobacteria, and Bacteroidetes (Fig. 2). Despite differences in sampling season, the bacterial taxa detected were consistent with previous studies on snow in the Arctic and Antarctic (Larose et al., 2010; Carpenter et al., 2000; Amato et al., 2007; Lopatina et al., 2013; Møller et al., 2013).

**Question 26**. Looking at Figure 4 there appears no real distinction between surface and sub-surface community composition. I wonder whether this relates to an arbitrary 15 cm distinction between surface and subsurface snow, rather than basing this on clearly defined snow layers?

**Response:**

We appreciate the reviewer to raise this concern. Figure 4 is the community composition at the phylum level, temporal changes were observed in certain phyla, namely the Alphaproteobacteria, Gammaproteobacteria, Firmicutes, Cyanobacteria, and Deinococcus-Thermus in the surface layer, and the Alphaproteobacteria, Firmicutes, Cyanobacteria, and Chloroflexi in the subsurface layer. This is largely consistent with those observed by Hell et al., 2013[1] that only a limited number of bacterial lineages exhibited changes across a short temporal scale in supraglacial snow.

1.  Hell, K., Edwards, A., Zarsky, J. et al. The dynamic bacterial communities of a melting High Arctic glacier snowpack. ISME J 7, 1814–1826 (2013)

**Question 27.** Same comments as above for the phraseology surrounding the data in figure S6 – there is a very large reliance here on simplistic correlations.

**Response:**

We appreciate the reviewer for raising this concern. We have performed additional linear regression analyses, and rephrased the sentence to reflect this.

Original manuscript:

In the surface layer, the relative abundances of Alphaproteobacteria, Gammaproteobacteria, and Firmicutes significantly decreased with time (Supplementary Fig S6), while those of Cyanobacteria and Deinococcus-Thermus significantly increased (all $P < 0.05$).

Amended manuscript (Lines 238-256):

The surface and subsurface snow were both dominated by Alphaproteobacteria, Actinobacteria, Cyanobacteria, Gammaproteobacteria, Bacteroidetes, Firmicutes, Chloroflexi, Gemmatimonadetes, Planctomycetes, Acidobacteria, Deltaproteobacteria, and Deinococcus-Thermus (Fig. 2). The relative abundance of most of these phyla was

not significantly differed in the two snow layers, except the Gemmatimonadetes, Planctomycetes, and Acidobacteria, which exhibited significantly higher relative abundance in the surface layer than in the subsurface layer (all $P < 0.05$, Wilcoxon rank-sum test; Supplementary Fig. S5). In the surface layer, negative associations were apparent in the relative abundances and ASV number of Alphaproteobacteria, Gammaproteobacteria, and Firmicutes with time ($F_{1,16} = 6.97$, $P = 0.018$, $R^2 = 0.30$; $F_{1,16} = 23.8$, $P < 0.001$, $R^2 = 0.60$, and $F_{1,16} = 22.28$, $P < 0.001$, $R^2 = 0.58$ in relative abundance; $F_{1,16} = 7.56$, $P = 0.014$, $R^2 = 0.32$; $F_{1,16} = 27.12$, $P < 0.001$, $R^2 = 0.63$, and $F_{1,16} = 16.68$, $P = 0.001$, $R^2 = 0.51$ in ASV number, respectively), while positive associations were apparent in the relative abundances and ASV number of Cyanobacteria and Deinococcus-Thermus with time ($F_{1,16} = 6.94$, $P = 0.018$, $R^2 = 0.30$ and $F_{1,16} = 13.10$, $P = 0.002$, $R^2 = 0.45$ in relative abundance; $F_{1,16} = 3.42$, $P = 0.083$, $R^2 = 0.18$ and $F_{1,16} = 4.07$, $P = 0.061$, $R^2 = 0.20$ in ASV number, respectively; Supplementary Fig. S6). In the subsurface layer, negative associations were apparent in the relative abundance and ASV number of Alphaproteobacteria and Firmicutes with time ($F_{1,16} = 15.17$, $P = 0.001$, $R^2 = 0.49$ and $F_{1,16} = 15.43$, $P = 0.001$, $R^2 = 0.49$ in relative abundance; $F_{1,16} = 18.98$, $P = 0.083$, $R^2 = 0.54$ and $F_{1,16} = 15.17$, $P = 0.001$, $R^2 = 0.53$ in ASV number, respectively, Supplementary Fig. S7), while positive associations were apparent in the relative abundance and ASV number of Cyanobacteria and Chloroflexi with time ($F_{1,16} = 5.62$, $P = 0.031$, $R^2 = 0.26$ and $F_{1,16} = 12.81$, $P = 0.003$, $R^2 = 0.44$ in relative abundance; $F_{1,16} = 5.34$, $P = 0.034$, $R^2 = 0.25$ and $F_{1,16} = 14.49$, $P = 0.002$, $R^2 = 0.47$ in ASV number, respectively).

**Section 3.3.**

**Question 28.** Similar comments to previous sections, in that much is made of relatively weak analysis outcomes (e.g. PCoA axis 1 = 16.31% of variation, and PCoA axis 2 = 11.06%), numerous correlations are employed that whilst they have 'significant' p-values often have very low r values, and nitrogen cycling gene content relative abundance that is inferred from 16S data rather than being measured on the samples. Given the lack of causative relationships established in these results care needs to be

taken on their interpretation regarding larger scale processes.

**Response:**

We appreciate the reviewer for raising this comment. We have added the regression analyses to evaluate the changes in bacterial diversity and environmental factors across the nine days. We agree that functions inferred from the 16S rRNA gene do not reflect the actual gene abundance changes, additional metagenomic and transcriptomic analyses would be required. This major drawback is discussed in the manuscript. We have also modified the discussion sections and the manuscript now focuses on the temporal patterns of bacterial diversity and environmental factors, and then carefully suggested that the nitrogen changes could be associated with the bacterial diversity changes. The results of functional gene predictions are only discussed to provide an alternative source of nitrogen in the surface snow and an alternative emission route in the subsurface snow.

Amended results:

On physicochemical properties (Lines 224-228):

[revised manuscript text omitted]

**Discussion:**

**Question 29.** Generally I find the direction of the discussion immediately and dominantly toward nitrogen dynamics to be concerning given that i) the nitrogen dataset is very modest (triplicate samples on a total of 7 days all within a single 9 day period) and ii) there remain a few concerns that I have detailed above about the validity of this dataset in terms of both analytics and sampling design. Much of the discussion text around the impact of nitrogen on the bacterial community is rather vague and is mainly conjecture. There are inappropriate references to non-snow studies (e.g. sub-glacial

works) and there is no attempt to place the findings into the wider context in regard to when/where samples were taken. For example, all samples are taken over a 9-day period in relatively recently deposited snow long before any melt processes are likely to begin – how representative are these data of what is actually happening in the snowpack on timescales longer than a week or so? Or how might they contrast to a melting snowpack? How do they fit with snowpack metamorphism processes?

**Response:**

We thank for the comment. We agree that we don't have solid evidence that the changes in bacterial diversity, community structure, and potential functions are caused by nitrogen. Instead, this conclusion was inferred from the correlation relationship. We have rewritten the discussion and focus primarily on the bacterial community and soil physiochemical property changes observed, and link the two very carefully. We have added a paragraph to discuss the implication of our short temporal scale study on the dynamics of bacterial and physicochemical properties changes in supraglacial snow. We have also modified to introduction and discussion sections to change the focus of the manuscript to the bacterial and nutrient dynamics across a short temporal scale, which has been largely omitted in the literature.

Amended manuscript

Introduction (Lines 50-63):

[revised manuscript text omitted]

**Question 30.** I believe that the first two sections as they stand could be reduced down into one short discussion paragraph on potential links between the bacterial community and nitrogen dynamics, but that the breadth of the dataset presented does not warrant the drawing of such definite conclusions on the controlling role of nitrogen within the snowpack sampled. The latter third paragraph is better and could form the basis for a new discussion of the dataset. The authors should provide a more general discussion into their datasets if this work is to be accepted for final publication. They should focus on what they can report rather than drawing grand conclusions from a relatively limited study and they should attempt to justify what insight their samples can provide given the timing and location of their sampling regime.

**Response:**

We thank the reviewer for the constructive suggestion. We have rewritten the second part of the discussion, which now discusses the potential sources of nitrogen (including nitrate and ammonium) in the surface snow, and their emission route in the subsurface snow. Traditionally, nitrogen deposition is responsible for the increase of nitrogen in surface snow, and here we calculated the theoretical input of nitrogen and showed the gap between this theoretical value and the observed value, and then propose bacterial nitrogen fixation could fill the gap. We also added a paragraph to discuss the implication of the present study, which showed a dynamic bacterial community in freshly fallen snow and the potential influence on nutrient transformation in supraglacial snow.

Both ammonium and nitrate concentrations increased in the surface snow (Fig. 1). The increase in ammonium is traditionally explained by biogenic emissions due to local vegetal and animal sources (Filippa et al., 2010), while the increase in nitrate has been largely attributed to atmospheric deposition (Björkman et al., 2014). Nitrogen deposition occurs at a rate of 282 kg N $km^{-2}$ $yr^{-1}$ in the region of our investigation (Lü and Tian, 2007), this equals 0.19 mg N for the 0.5 m × 0.5 m area sampled each day (assuming nitrogen deposition occurred evenly across the year). If further assuming the deposited nitrogen only affects the surface snow (i.e., the top 15 cm as defined in the present study), the daily nitrogen increase is estimated to be 0.084 mg N $L^{-1}$. This is lower than the slope of total nitrogen increase observed in the surface snow of the present study (0.21 mg N $L^{-1}$ $day^{-1}$). Thus, either the atmospheric nitrogen deposition has more than doubled, or bacterial nitrogen fixation could be an alternative source of nitrogen input (Telling et al., 2011). The latter is supported by the biosynthesis of nitrogen-containing compounds by bacteria with increased dissolved organic nitrogen reported in the Antarctic surface snow (Antony et al., 2017). The contribution of bacterial nitrogen fixation is further supported by the increase in the relative abundance of Cyanobacteria and the predicted abundance of *nifH* gene in surface snow (Supplementary Fig. S6 and Fig. S12). The exact nitrogen fixation rate was not quantified in the present study, but the results suggest that microbial nitrogen fixation could be an overlooked source of nitrogen in Tibetan glacier snow, further transcriptomic and nitrogen-isotope analyses may provide further evidence on the microbial activity in nitrogen fixation.

In contrast with the surface layer, nitrogen concentrations (nitrate and ammonium) significantly decreased in the subsurface snow with time (Fig. 1). In a snow reactive nitrogen oxides ($NO_y$) survey in Greenland, $NO_y$ flux was reported to exit snow in 52 out of 112 measurements, and the magnitude cannot be explained by the photolysis of nitrate alone (Dibb et al., 1998). Furthermore, the short sampling period of the present study does not allow rapid photolysis to occur (Larose et al., 2013b), therefore collectively suggesting an alternative source of $NO_y$ emission could exist. The

denitrification process could contribute to nitrogen consumption, which is evidenced by the increase of predicted genes associated with denitrification processes (*narG*; Supplementary Fig. S12) (Telling et al., 2011; Zhang et al., 2020). This is consistent with the high relative abundance of denitrification-related genes being detected in the snowpack of Spitsbergen Island of Svalbard, Norway (Larose et al., 2013a). Despite the oxygen level in the subsurface snow was not measured, the occurrence of anaerobic denitrification reactions in subsurface snow has been reported in Arctic snowpacks (Larose et al., 2013a). Furthermore, Poniecka et al. (2018) showed that cryoconite microorganisms can generate an anoxic zone 2 mm below the sediment surface within an hour. Thus, anaerobic pockets in subsurface snow at 15-30 cm deep could exist, which allows denitrification reactions to occur. Further metatranscriptomic analyses targeting the genes associated with nitrogen cycling are required to further confirm the distinct nitrogen transformation processes between the surface and subsurface layers.

---

## Author Comment (AC2)

**We thank Anonymous Referee #2 for their considered, detailed, and helpful review of our manuscript. Below we present our point-by-point responses.**

**L77-79:**

1) At each time point, three replicate samples were collected within a very small area of only 5 m by 3 m. The authors do not state clearly how far apart each replicate was, only that they were at least 30 cm apart. This introduces several serious issues:

**Question 1.** a) The size of the ablation zone is not given in the manuscript, but for most glaciers, the ablation zone would be a lot larger than 5x3 m, and by sampling only within a very small portion of that zone, it is open to discussion whether those samples are representative of the whole ablation zone.

**Response:**

We thank reviewer #2 for raising this concern. We did not intend to represent the entire ablation zone, instead, we attempted to focus on the nutrient and bacterial community dynamics immediately following a snowfall event, which will be buried following the next precipitation. Furthermore, we did not sample from the entire ablation zone to avoid the influence of spatial heterogeneity. The introduction, method, and the title were modified to represent the scope of the present in a better way.

Amended manuscript (Title):

Temporal variation of bacterial community and nutrients in Tibetan glacier snowpack

Introduction (Lines 50-63):

Several studies investigated the dynamics of nutrient and bacterial changes in supraglacial snow during the ablation period. Larose et al. (2013a) revealed that the form of nitrogen varied as a function of time in supraglacial snow during a two-month field study at the Svalbard, Norway and fluctuations in microbial community structure have been reported with the relative abundance of fungi and bacteria (such as Bacteroidetes and Proteobacteria) increased and decreased, relatively. Seasonal shifts in snowpack bacterial communities have also been reported in the mountain snow in Japan, rapid microbial growth was observed with increasing snow temperature and

meltwater content (Segawa et al., 2005). However, the results of these studies are likely the consequence of these several precipitation events due to the long period of time. During precipitation, a new snow layer forms above the previous ones, which is responsible for the stratified snowpack structure. These different snow layers have distinct physical and chemical characteristics and their age also differed substantially (Lazzaro et al., 2015). Thus, the microbial process across the aged snowpack could be complex, whereas focusing on supraglacial snow from a single snowfall event could provide unique insights into the bacterial and nutrient dynamics. Hell et al. (2013) reported bacterial community structure changes during the ablation period across five days in the high Arctic, while the bacterial and nutrient dynamics during the snow accumulation period remain elusive.

And (Lines 83-93):

Glacier melting increases the discharge of microorganisms and nutrients in meltwater into downstream aquatic ecosystems (Kohler et al., 2020), which substantially impacts the bacterial community and biogeochemical processes (Liu et al., 2021). Thus, it is crucial to understand the transformation processes of the bacterial community and nutrients in the supraglacial snow. Several studies have investigated the nutrient and bacterial community changes in supraglacial snow across the winter (Brooks et al., 1998; Liu et al., 2006), but the bacterial and nutrient dynamics of freshly fallen snow have been largely overlooked. These short temporal changes will influence the following post-depositional processes after it is buried by the next snowfall, and will ultimately determine the physicochemical properties of the stratified snow in the following year. In the present study, we investigated the bacterial community and snow physiochemical property changes in the surface and subsurface supraglacial snow during a nine-day period after a single snowfall event at the Dunde Glacier on the northeast of the Tibetan Plateau.

**Question 2.** b) Since the three replicates were collected so close to each other, they could be considered technical replicates, rather than true replicates, which would have

a knock-on effect on what statistical tests would be suitable.

**Response:**

Unfortunately, we do not fully agree that the three samples of the same day are technical replicates. By definition, technical replicates are repeated measurements of the same sample. The three snowpits were dug randomly within the 5 m × 3 m area, and any two snow pits were at least 30 cm apart. This sampling design is to ensure that the sample differences are not caused by heterogeneity from a large geospatial distance, while we are not sampling two adjacent areas on the same day (as the reviewer has suggested) to avoid the results observed to be caused by geospatial pattern.

**Question 3.** c) The snow pits could have been either 30 cm apart or even 4.5 meters apart, that is a huge difference and without a sampling strategy that takes spatial variation into consideration, it is impossible for the reader to know whether the differences seen are due to spatial or temporal variation – for instance, is this perhaps the reason why the communities are very different on some days but very similar on others? Therefore, the spatial distribution of each individual snow pit and distances between pits must be clearly presented in the manuscript.

**Response:**

We chose a small flat area for sampling to reduce the impact of spatial variation in snowfall. The distance between the three was were mostly similar, generally between 30-50 cm apart. This is mainly due to a footpath that needs to be left between snowpits, and samples were not taken on the area where footprint was left. Given the short distance among the three snowpits and the exact distance between snowpits was not kept, we cannot include this distance in the calculation. In addition, the differences in the bacterial community and physiochemical factors among the three snowpits of the same day are most likely random. We have modified the description in the sampling design to clarify this.

Amended manuscript (Lines 105-124):

No supraglacial snow was observed on the glacier surface on the 10th of October when first arrived at the camp. Snowfall started on the 18th and ended on the 23rd of October.

Sampling was conducted over a nine-day period after the snowfall stopped on a flat 5 m × 3 m small area to reduce the impact of sample heterogeneity due to spatial variations. Snow samples were collected on the 24th, 25th, 26th, 27th, and 29th of October, and the 2nd November (which are referred as day 1, 2, 3, 4, 6, and 9) until the next snowfall started. This enabled us to follow the development of bacterial communities and the chemical environment through time after deposition. The ambient air temperature at the sampling period is averaged -8 °C (data available through the European Centre for Medium-Range Weather Forecasts, Supplementary Fig. S2), no snow melting was observed over the nine days.

On each day, three snow pits were randomly dug within the 5 m × 3 m area and any two snow pits were 30-50 cm apart. Each snow pit was approximately 30 cm deep, then the snow was further divided equally into the surface and subsurface layers (approximately 15 cm deep for each layer) to get enough snow samples to extract DNA, after Carey et al. (2016). For each snow pit, the top 1 cm in contact with the air was removed using a sterile spoon to avoid contamination, and then surface and subsurface snow were collected using a sterilized Teflon shovel into 3 L sterile sampling bags separately. Approximately 100 mL were used for physicochemical analyses, whereas the rest was used for DNA extraction. A total of 36 samples were collected. Tyvek bodysuits and latex gloves were worn during the entire sampling process to minimize the potential for contamination, and gloves were worn during all subsequent handling of samples. Samples were kept frozen during the transportation to the laboratory and stored at -20 °C until analysis.

**Question 4.** If the distances between snowpits varied substantially between sampling days, I also suggest that the authors include distance as a factor in their statistical analyses.

**Response:**

We thank the reviewer for this constructive suggestion. The distance between the three snowpits was mostly similar, generally between 30-50 cm apart. As the exact distance between snowpits was not kept, we cannot include the distance between the three

snowpits as a spatial factor in the analysis. Nevertheless, the differences in the bacterial community and physiochemical factors among the three snowpits of the same are most likely random. Therefore, the distance is unlikely to have a substantial impact on the diversity and community patterns, we have modified the description in sampling design to clarify this.

Amended manuscript (Lines 105-124):

No supraglacial snow was observed on the glacier surface on the 10[th] of October when first arrived at the camp. Snowfall started on the 18[th] and ended on the 23[rd] of October. Sampling was conducted over a nine-day period after the snowfall stopped on a flat 5 m × 3 m small area to reduce the impact of sample heterogeneity due to spatial variations. Snow samples were collected on the 24[th], 25[th], 26[th], 27[th], and 29[th] of October, and the 2[nd] November (which are referred as day 1, 2, 3, 4, 6, and 9) until the next snowfall started. This enabled us to follow the development of bacterial communities and the chemical environment through time after deposition. The ambient air temperature at the sampling period is averaged -8 °C (data available through the European Centre for Medium-Range Weather Forecasts, Supplementary Fig. S2), no snow melting was observed over the nine days.

On each day, three snow pits were randomly dug within the 5 m × 3 m area and any two snow pits were 30-50 cm apart. Each snow pit was approximately 30 cm deep, then the snow was further divided equally into the surface and subsurface layers (approximately 15 cm deep for each layer) to get enough snow samples to extract DNA, after Carey et al. (2016). For each snow pit, the top 1 cm in contact with the air was removed using a sterile spoon to avoid contamination, and then surface and subsurface snow were collected using a sterilized Teflon shovel into 3 L sterile sampling bags separately. Approximately 100 mL were used for physicochemical analyses, whereas the rest was used for DNA extraction. A total of 36 samples were collected. Tyvek bodysuits and latex gloves were worn during the entire sampling process to minimize the potential for contamination, and gloves were worn during all subsequent handling of samples. Samples were kept frozen during the transportation to the laboratory and stored at -20 °C until analysis.

**Question 5.** 2) It is not clear from the Methods, whether snowpits dug on subsequent days were from within the same 5x3 m square or if a new sampling area was selected each time? Regardless, this needs to be clarified in the manuscript, but, if the former, the data ought to be analysed as a time series and not as a series of independent sampling points.

**Response:**

These snow pits were dug within the same 5*3 m square. We agree that time-series analyses would be required. Therefore, we have performed additional linear regression analyses on the changes in bacterial community diversity and geochemical properties.

Amended manuscript (Lines 107-110):

Snow samples were collected on the 24th, 25th, 26th, 27th, and 29th of October, and the 2nd November (which are referred as day 1, 2, 3, 4, 6, and 9) until the next snowfall started.

And (Lines 114-115):

On each day, three snow pits were randomly dug within the 5 m × 3 m area and any two snow pits were 30-50 cm apart.

**Question 6.** L101: The filtration setup and method for filtering need to be described in more detail, especially with regards to what steps were taken to prevent contamination, how the samples were thawed and the time scales involved.

**Response:**

We thank for this comment. We have added the filtration setup and procedures into the method section.

Original manuscript:

For assessing the bacterial community composition, each of the melted snow samples (3 L) was filtered onto a 0.22 μm polycarbonate membrane (Millipore, USA) with a vacuum pump (Ntengwe 2005).

Amended manuscript

The 100 mL snow sample for physicochemical analysis was melted at room temperature for 3 hours before being analysed. For dissolved organic carbon (DOC) and major ions measurements, 100 mL of snow meltwater was syringe-filtered through a 0.45 μm polytetrafluoroethylene (PTFE) membrane filter (Macherey–Nagel) into 20-mL glass bottles. The membrane has been pre-treated with 1% HCl, deionized water rinsed, and 450 °C > 3 h combusted to remove any potential carbon and nitrogen on the membrane, and the initial 10 mL of the filtrate was discarded before collecting the sample for analysis to eliminate any residual compound on the membrane.

For assessing the bacterial community composition, snow samples (3 L) were melted at 4 °C overnight and filtered onto a sterile 0.22 μm polycarbonate membrane (Millipore, USA) with a vacuum pump (Ntengwe 2005).

**Question 7.** L149-154: Although functional profiling of taxa identified by 16S sequencing can give some insights into what abilities the community might have, it is not suitable for this kind of investigation. Although shotgun metagenome sequencing may be out of scope, the authors should be able to provide much more accurate and quantitative data for the presence of nitrogen fixers and denitrifiers in their samples by qPCR of relevant genes (e.g. nifH and narG). In its current form, the method used does not provide the data needed to back up the conclusions drawn by the authors.

**Response:**

We totally agree with the reviewer that functional prediction based on 16S rRNA gene data is not fully reliable, despite the original publication of *Tax4Fun2* package revealing a high correlation between the function predicted and those from metagenome data. More accurate and quantitative measurements (including metagenome sequencing and qPCR) are necessary to confirm the results. Unfortunately, the concentration of the sample was quite low and they were barely enough for amplicon sequencing. The functional prediction was used to explain the nitrogen (nitrate and ammonium) changes. Instead to be a definitive explanation, the pattern of changes in these nitrogen cyclingrelated genes provides a hint on the potential function changes. We have amended the manuscript so the discussion of functional prediction results was minimized, and it was used to propose an alternative route of nitrogen accumulation and consumption in the surface and subsurface snow.

Amended manuscript (Lines 364-406):

Both ammonium and nitrate concentrations increased in the surface snow (Fig. 1). The increase in ammonium is traditionally explained by biogenic emissions due to local vegetal and animal sources (Filippa et al., 2010), while the increase in nitrate has been largely attributed to atmospheric deposition (Björkman et al., 2014). Nitrogen deposition occurs at a rate of 282 kg N km$^{-2}$ yr$^{-1}$ in the region of our investigation (Lü and Tian, 2007), this equals 0.19 mg N for the 0.5 m $\times$ 0.5 m area sampled each day (assuming nitrogen deposition occurred evenly across the year). If further assuming the deposited nitrogen only affects the surface snow (i.e., the top 15 cm as defined in the present study), the daily nitrogen increase is estimated to be 0.084 mg N L$^{-1}$. This is lower than the slope of total nitrogen increase observed in the surface snow of the present study (0.21 mg N L$^{-1}$ day$^{-1}$). Thus, either the atmospheric nitrogen deposition has more than doubled, or bacterial nitrogen fixation could be an alternative source of nitrogen input (Telling et al., 2011). The latter is supported by the biosynthesis of nitrogen-containing compounds by bacteria with increased dissolved organic nitrogen reported in the Antarctic surface snow (Antony et al., 2017). The contribution of bacterial nitrogen fixation is further supported by the increase in the relative abundance of Cyanobacteria and the predicted abundance of *nifH* gene in surface snow (Supplementary Fig. S6 and Fig. S12). The exact nitrogen fixation rate was not quantified in the present study, but the results suggest that microbial nitrogen fixation could be an overlooked source of nitrogen in Tibetan glacier snow, further transcriptomic and nitrogen-isotope analyses may provide further evidence on the microbial activity in nitrogen fixation.

In contrast with the surface layer, nitrogen concentrations (nitrate and ammonium) significantly decreased in the subsurface snow with time (Fig. 1). In a snow reactive nitrogen oxides (NO$_y$) survey in Greenland, NO$_y$ flux was reported to exit snow in 52

out of 112 measurements, and the magnitude cannot be explained by the photolysis of nitrate alone (Dibb et al., 1998). Furthermore, the short sampling period of the present study does not allow rapid photolysis to occur (Larose et al., 2013b), therefore collectively suggesting an alternative source of $NO_y$ emission could exist. The denitrification process could contribute to nitrogen consumption, which is evidenced by the increase of predicted genes associated with denitrification processes (*narG*; Supplementary Fig. S12) (Telling et al., 2011; Zhang et al., 2020). This is consistent with the high relative abundance of denitrification-related genes being detected in the snowpack of Spitsbergen Island of Svalbard, Norway (Larose et al., 2013a). Despite the oxygen level in the subsurface snow was not measured, the occurrence of anaerobic denitrification reactions in subsurface snow has been reported in Arctic snowpacks (Larose et al., 2013a). Furthermore, Poniecka et al. (2018) showed that cryoconite microorganisms can generate an anoxic zone 2 mm below the sediment surface within an hour. Thus, anaerobic pockets in subsurface snow at 15-30 cm deep could exist, which allows denitrification reactions to occur. Further metatranscriptomic analyses targeting the genes associated with nitrogen cycling are required to further confirm the distinct nitrogen transformation processes between the surface and subsurface layers.

**Minor points:**

**Question 8.** L19 and elsewhere: When contrasting data for the surface and subsurface layers, it is easier for the reader to follow if data and location is grouped together, e.g. "Nitrate and ammonium concentrations increased in the surface and decreased in the subsurface snow over time, therefore indicating accumulation and consumption processes, respectively." The same goes for the following sentence re nitrogen fixation and denitrification genes.

**Response:**

We appreciate the reviewer for this comment. We have rewritten the sentence to make the data and location grouped.

Original manuscript:

Nitrate and ammonium concentration increased and decreased in the surface and

subsurface snow over time, therefore indicating accumulation and consumption processes, respectively. This is also evidenced by the dominance of organisms predicted to carry nitrogen fixation and denitrification genes in the surface and subsurface layers, respectively.

Amended manuscript (Lines 21-26):

Nitrate and ammonium concentrations increased from 0.44 to 1.15 mg/L and 0.18 to 0.24 mg/L in the surface snow and decreased from 3.81 to 1.04 mg/L and 0.53 to 0.25 mg/L in the subsurface snow over time, therefore indicating accumulation and consumption processes, respectively. The nitrate concentration covaried with bacterial diversity, community structure, and the predicted nitrogen fixation and denitrification-related genes, suggesting nitrogen could mediate bacterial community changes.

**Question 9.** Introduction: I though the Introduction was particularly well-written and set the scene very well.

**Response:**

We greatly thank reviewer #2 for this encouragement.

**Question 10.** L71: Delete the "the" in front of "October".

**Response:**

The grammar mistake has been corrected as indicated by the reviewer.

Original manuscript:

Snow samples were collected from the ablation zone at Dunde glacier (38°06′N, 96°24′E, 5325 m above the sea-level), during the October and November, 2016 (Supplementary Fig S1).

Amended manuscript (Lines 102-103):

Snow samples were collected from the ablation zone at Dunde glacier (38°06′N, 96°24′E, 5325 m above the sea-level), during October and November, 2016 (Supplementary Fig. S1).

**Question 11.** L74: There is a mismatch between the dates and the number of dates. Day

5 (28th of October) is missing from the list of Dates and the 2nd of November ought to be day 10.

**Response:**

We apologize for the mistake. The mismatch mistake has been corrected.

Original manuscript:

Sampling was conducted over a nine-day period (on the $24^{th}$, $25^{th}$, $26^{th}$, $27^{th}$, and $29^{th}$ of October, and the $2^{nd}$ November, which are referred as day 1, 2, 3, 4, 5, 6, and 9 thereafter).

Amended manuscript (Lines 107-110):

Snow samples were collected on the $24^{th}$, $25^{th}$, $26^{th}$, $27^{th}$, and $29^{th}$ of October, and the $2^{nd}$ November (which are referred as day 1, 2, 3, 4, 6, and 9) until the next snowfall started.

**Question 12.** L75: What is the age of the snowpack? I.e. when did the snow first start accumulating in this area?

**Response:**

The snowpack was freshly formed. The surface of the glacier was icy when first arrived. Snowfall started on the $18^{th}$ of October and stopped on the $23^{rd}$ of October. Sampling started on the $24^{th}$ of October. We have amended the method to reflect this.

Amended manuscript (method section) (Lines 105-113):

No supraglacial snow was observed on the glacier surface on the $10^{th}$ of October when first arrived at the camp. Snowfall started on the $18^{th}$ and ended on the $23^{rd}$ of October. Sampling was conducted over a nine-day period after the snowfall stopped on a flat 5 m × 3 m small area to reduce the impact of sample heterogeneity due to spatial variations. Snow samples were collected on the $24^{th}$, $25^{th}$, $26^{th}$, $27^{th}$, and $29^{th}$ of October, and the $2^{nd}$ November (which are referred as day 1, 2, 3, 4, 6, and 9) until the next snowfall started. This enabled us to follow the development of bacterial communities and the chemical environment through time after deposition. The ambient air temperature at the sampling period is averaged -8 °C (data available through the European Centre for Medium-Range Weather Forecasts, Supplementary Fig. S2), no

snow melting was observed over the nine days.

**Question 13.** L89: How were the 0.45 um cellulose membrane filters treated before sampling? Was the initial volume of filtrate discarded before collecting the sample for analysis?

**Response:**

The 0.45 μm polytetrafluoroethylene (PTFE) membrane filter was treated with 1% HCl, deionized water rinsed, and then incubated at 450 °C for 3 hours to eliminate carbon or nitrogen contamination from the filters. The first 10 ml of the filtrate was discarded before collecting the sample for analysis. We added the following sentences in the method part.

Amended manuscript (Lines 126-131):

The 100 mL snow sample for physicochemical analysis was melted at room temperature for 3 hours before being analysed. For dissolved organic carbon (DOC) and major ions measurements, 100 mL of snow meltwater was syringe-filtered through a 0.45 μm polytetrafluoroethylene (PTFE) membrane filter (Macherey–Nagel) into 20-mL glass bottles. The membrane has been pre-treated with 1% HCl, deionized water rinsed, and 450 °C > 3 h combusted to remove any potential carbon and nitrogen on the membrane, and the initial 10 mL of the filtrate was discarded before collecting the sample for analysis to eliminate any residual compound on the membrane.

**Question 14.** Section 2.7 Statistical analysis (and elsewhere): Function and package names need to be consistently highlighted (single and double quotation marks and no highlighting at all are all used with no apparent system to it). Usually, function and package names are italicised, but any consistent form of highlighting would work.

**Response:**

We appreciate the reviewer for this suggestion. We have rewritten the "Statistical analysis" part as below.

Amended manuscript (Lines 189-213):

Shannon-Wiener and Chao1 indices, which were used to estimate the species richness

in the snow community, were calculated using the "*diversity*" function in the R package "*vegan*" (Oksanen et al., 2010). Functional profiling of bacterial taxa was carried out using the package "*Tax4Fun2*" in R (Wemheuer et al., 2020). While the application of functional profiles predicted from 16S rRNA gene-based community composition data is limited by the functional information available in databases, we present these data as one possible interpretation of the patterns we detected, and note that the "*Tax4Fun2*" package performed well compared to older widely used programs (Wemheuer et al., 2020). The pairwise Wilcoxon rank-sum test was used to compare the depth-horizon differences in environmental variables, alpha-diversity, and the relative abundance of taxonomic groups at the phylum level. Linear regression modelling was implemented in R using the "*lm*" function to estimate the trend of changes over time. The bacterial community structure was subjected to principal coordinate analysis (PCoA) carried out using the "*pcoa*" function of the "*ape*" package in R. The significance of dissimilarity of community composition among samples was tested using permutational multivariate analysis of variance (PERMANOVA) based on Bray-Curtis distance metrics with the "*adonis*" function in the R package "*vegan*" (Oksanen et al., 2010). Test results with $P < 0.05$ were considered statistically significant. Mantel test based on Spearman's rank correlations was performed using the bacterial dissimilarity and environmental dissimilarity matrix, calculated based on the Bray-Curtis distance metrics and Euclidean distance metrics in the "*vegan*" R package, respectively. The normalized stochasticity ratio (NST) based on the Bray–Curtis dissimilarity was calculated using the "*NST*" package in R to estimate the determinacy and stochasticity of the bacterial assembly processes with high accuracy and precision (Ning et al., 2019). The NST index used 50% as the boundary point between more deterministic (<50%) and more stochastic (>50%) assembly processes. All environmental variables were normalized before the calculation. All statistical analyses were executed in R version 3.4.3 (R Core Team, 2017).

**Question 15.** L195-197 and L203-204: These to statements are contradicting each other. Either there was no significant difference in relative abundance between the two layers

or there was a significant difference in bacterial community structure between the two layers.

**Response:**

We apologize for not stating this clear enough. The non-significant difference was for the community structure at the phylum level (line 195-197), whereas the significant difference was observed at the ASV level (line 203-204). We have rephrased the sentence to clarify this.

Amended manuscript (Lines 274-276):

The bacterial community structure at the ASV level significantly differed in the surface and subsurface snow (PERMANOVA, F = 2.78, $P < 0.001$, Fig. 5a), as well as among the different sampling times (PERMANOVA, F = 3.31, $P < 0.001$ and F = 2.17, $P < 0.001$, respectively).

**Question 16.** L198-201: It would help to back this up with absolute numbers of ASVs. E.g. did the cyanobacteria and Chloroflexi really grow in numbers in the subsurface layer over time or did their populations stay the same (in actual abundance) while those of the alpha-Proteobacteria and Firmicutes declined, thereby resulting in an apparent increase due to the increase in relative abundance?

**Response:**

We totally agree with the reviewer on this. The increase in relative abundance could be either due to the increased Cyanobacteria abundance or the reduction of non-Cyanobacteria bacteria. However, this is one of the drawbacks of amplicon sequencing that the absolute number reads depends not only on the abundance of organisms but also on the depth of sequencing. Nevertheless, we plotted the ASV number of Cyanobacteria and Chloroflexi, which also increased in the subsurface layer over time.

Original manuscript:

In the surface layer, the relative abundances of Alphaproteobacteria, Gammaproteobacteria, and Firmicutes significantly decreased with time (Supplementary Fig S6), while those of Cyanobacteria and Deinococcus-Thermus significantly increased (all $P < 0.05$). In the subsurface layer, the relative abundance of

Alphaproteobacteria and Firmicutes significantly decreased with time, while Cyanobacteria and Chloroflexi significantly increased (all $P < 0.05$).

Amended manuscript (Lines 243-256):

In the surface layer, negative associations were apparent in the relative abundances and ASV number of Alphaproteobacteria, Gammaproteobacteria, and Firmicutes with time ($F_{1,16} = 6.97$, $P = 0.018$, $R^2 = 0.30$; $F_{1,16} = 23.8$, $P < 0.001$, $R^2 = 0.60$, and $F_{1,16} = 22.28$, $P < 0.001$, $R^2 = 0.58$ in relative abundance; $F_{1,16} = 7.56$, $P = 0.014$, $R^2 = 0.32$; $F_{1,16} = 27.12$, $P < 0.001$, $R^2 = 0.63$, and $F_{1,16} = 16.68$, $P = 0.001$, $R^2 = 0.51$ in ASV number, respectively), while positive associations were apparent in the relative abundances and ASV number of Cyanobacteria and Deinococcus-Thermus with time ($F_{1,16} = 6.94$, $P = 0.018$, $R^2 = 0.30$ and $F_{1,16} = 13.10$, $P = 0.002$, $R^2 = 0.45$ in relative abundance; $F_{1,16} = 3.42$, $P = 0.083$, $R^2 = 0.18$ and $F_{1,16} = 4.07$, $P = 0.061$, $R^2 = 0.20$ in ASV number, respectively; Supplementary Fig. S6). In the subsurface layer, negative associations were apparent in the relative abundance and ASV number of Alphaproteobacteria and Firmicutes with time ($F_{1,16} = 15.17$, $P = 0.001$, $R^2 = 0.49$ and $F_{1,16} = 15.43$, $P = 0.001$, $R^2 = 0.49$ in relative abundance; $F_{1,16} = 18.98$, $P = 0.083$, $R^2 = 0.54$ and $F_{1,16} = 15.17$, $P = 0.001$, $R^2 = 0.53$ in ASV number, respectively, Supplementary Fig. S7), while positive associations were apparent in the relative abundance and ASV number of Cyanobacteria and Chloroflexi with time ($F_{1,16} = 5.62$, $P = 0.031$, $R^2 = 0.26$ and $F_{1,16} = 12.81$, $P = 0.003$, $R^2 = 0.44$ in relative abundance; $F_{1,16} = 5.34$, $P = 0.034$, $R^2 = 0.25$ and $F_{1,16} = 14.49$, $P = 0.002$, $R^2 = 0.47$ in ASV number, respectively).

[Figure]

Fig S7 Temporal changes of the ASV number of dominant bacterial phyla in the surface and subsurface snow. Each dot represents an individual sample. The solid and dashed lines indicate significant and nonsignificant changes, respectively. Significance is based on linear regression.

**Question 17.** L206-210: It would be useful to see how the environmental factors correlate with these axes.

**Response:**

We performed linear regression for the environmental factors against the PCoA axes. The result showed that the PCOA1 did not correlate with any environmental factors both in the surface and subsurface layers. However, the PCOA2 showed a significant correlation with all environmental factors except $Na^+$ in the subsurface layer. We have added this result to the manuscript.

Amended manuscript (Lines 278-284):

Specifically, only the second principal coordinate (PCoA2) values of the surface snow significantly varied with time ($F_{1,16} = 141.8$, $P < 0.001$, $R^2 = 0.89$, Fig. 5b), while the PCoA1 values of the surface snow did not. Furthermore, PCoA1 and PCoA2 of the surface snow exhibited no significant correlation with the measured environmental

factors (Supplementary Fig. S9 and S10). In comparison, both PCoA1 and PCoA2 values of the subsurface snow co-varied with time ($F_{1,16}$ = 6.35, $P$ = 0.023, $R^2$ = 0.28 and $F_{1,16}$ = 8.38, $P$ = 0.011, $R^2$ = 0.34, respectively, Fig. 5b), while the PCoA2 also demonstrated significant association with nitrate, ammonium, potassium, sulfate, and DOC concentrations (Supplementary Fig. S10).

[Figure]

Fig S9: Pairwise regression analyses between PCoA1 scores and environmental factors. The solid and dashed lines indicate significant and nonsignificant changes (based on linear regression at $P$ < 0.05), respectively. PCoA1 exhibits no significant relationship with the measured environmental factors in the surface snow, while in the subsurface layer, the PCoA1 is significantly associated with DOC concentrations.

[Figure]

Fig S10 Pairwise regression analyses between PCoA2 scores and environmental factors. The solid and dashed lines indicate significant and nonsignificant changes (based on linear regression at $P <$ 0.05), respectively. PCoA2 exhibits no significant relationship with the measured environmental factors in the surface layer, while in the subsurface layer, the PCoA2 is significantly associated with nitrate, ammonium, potassium, sulfate, and DOC concentrations.

**Question 18.** L245-246, 253: It would really help the authors argument here if they presented a theoretical input of nitrogen for each sample. Given a yearly deposition rate of 282 kg N per km2 and based on the seasonal deposition pattern for glaciers in the region, how much nitrogen would they have expected in the volume of snow that they collected for nitrogen analysis?

**Response:**

We appreciate the reviewer for this constructive suggestion. We have calculated the theoretical input of nitrogen for our samples, and this information was used to demonstrate the existence of a nitrogen input gap in the present study. We then proposed nitrogen fixation could be responsible.

Amended manuscript (Lines 368-386):

Nitrogen deposition occurs at a rate of 282 kg N $km^{-2}$ $yr^{-1}$ in the region of our investigation (Lü and Tian, 2007), this equals 0.19 mg N for the 0.5 m × 0.5 m area sampled each day (assuming nitrogen deposition occurred evenly across the year). If further assuming the deposited nitrogen only affects the surface snow (i.e., the top 15 cm as defined in the present study), the daily nitrogen increase is estimated to be 0.084 mg N $L^{-1}$. This is lower than the slope of total nitrogen increase observed in the surface snow of the present study (0.21 mg N $L^{-1}$ $day^{-1}$). Thus, either the atmospheric nitrogen deposition has more than doubled, or bacterial nitrogen fixation could be an alternative source of nitrogen input (Telling et al., 2011). The latter is supported by the biosynthesis of nitrogen-containing compounds by bacteria with increased dissolved organic nitrogen reported in the Antarctic surface snow (Antony et al., 2017). The contribution of bacterial nitrogen fixation is further supported by the increase in the relative abundance of Cyanobacteria and the predicted abundance of *nifH* gene in surface snow (Supplementary Fig. S6 and Fig. S12). The exact nitrogen fixation rate was not quantified in the present study, but the results suggest that microbial nitrogen fixation could be an overlooked source of nitrogen in Tibetan glacier snow, further transcriptomic and nitrogen-isotope analyses may provide further evidence on the microbial activity in nitrogen fixation.

**Question 19.** L255, 5 Conclusion: Since Tot-N is decreasing in the subsurface over time, the nitrogen is clearly not incorporated into biomass. It would be useful to see a brief discussion on how the authors think the nitrogen is leaving the system. The surface community may not be negatively impacted by increased N deposition, but would the subsurface community be able to cope with an increased N input or would it be exported downstream and add to the N load in glacier-fed rivers?

**Response:**

We appreciate the reviewer for this constructive suggestion. We have added a brief discussion in the conclusion section on the impact of enhanced nitrogen deposition on the subglacial ecosystem and downstream ecosystems.

Amended manuscript (Lines 458-466):

Due to atmospheric nitrogen deposition and bacterial nitrogen fixation activities, nitrogen limitation is unlikely to occur in the surface snow, thus additional nitrogen deposition due to global climate change is unlikely to substantially impact the bacterial community in surface snow. In contrast, nitrogen consumption was inferred in the subsurface snow. Nitrogen is traditionally recognized to be released from supraglacial environmental due to photolysis, whereas the present study hints that bacterial denitrification process could be an alternative route. Therefore, the increased nitrogen deposition due to anthropogenic activities may enhance the denitrification process in the subsurface snow. The enhanced nitrogen emission could reduce the impact of increased nitrogen deposition on downstream glacier-fed rivers, but may feedback global warming positively.

**Question 20.** L258: There is no mention of oxygen levels being measured at the of sampling in the manuscript. If the authors believe that the oxygen levels in the subsurface layer of the snow pack can be expected to be sufficiently low to allow for denitrification to occur based on data from the literature, that evidence needs to be presented in the manuscript. Regarding test for correlation: A weak correlation is still weak, even if the test is highly significant. Also, I am not clear on why the authors are using correlation tests to test for changes in environmental variables over time?

**Response:**

We appreciate the reviewer for this comment. We have added relevant references to the manuscript to evidence that denitrification could occur in the subsurface snow (i.e., 15-30 cm). We have also added linear regression analysis to support the temporal pattern of the bacterial diversity and environmental factors changes across the nine days.

Amended manuscript (Lines 397-406):

Despite the oxygen level in the subsurface snow was not measured, the occurrence of anaerobic denitrification reactions in subsurface snow has been reported in Arctic snowpacks (Larose et al., 2013a). Furthermore, Poniecka et al. (2018) showed that cryoconite microorganisms can generate an anoxic zone 2 mm below the sediment surface within an hour. Thus, anaerobic pockets in subsurface snow at 15-30 cm deep

could exist, which allows denitrification reactions to occur. Further metatranscriptomic analyses targeting the genes associated with nitrogen cycling are required to further confirm the distinct nitrogen transformation processes between the surface and subsurface layers.

**Question 21.** Figure 5: 1) Some labels are missing from markers in Fig 5a, 2) Out of curisosity: In most cases the samples taken on the same day are very closely clustered (e.g. Surface Day 1, 4 and 9; Subsurface Day 9), so what is special about Subsurface Day 3 and some of the others, where the replicate samples are very different from each other?

**Response:**

We apologize for the missing labels. We revised the figure as below. The large community variation in Days 3 and 6 could be due to the larger variation of nitrate in the snow sampled. This is likely to be random, such as a large chunk of dust could be in one of the samples collected on these days.

[Figure]

**Fig. 5 Principal coordinate analysis (PCoA) of microbial communities in the surface and subsurface snow.**

(a) Bray-Curtis distance-based PCoA ordination plot. The microbial community structures of the surface and subsurface snows are significantly different (PERMANOVA, $P < 0.001$). (b) Pairwise regression analysis between PCoA scores and sampling time. The solid and dashed lines indicate significant and insignificant changes (based on linear regression), respectively. The PCoA1

scores for the bacterial community in surface layer exhibit no significant correlation with time, while the PCoA2 scores significantly correlated with time. The PCoA1 and PCoA2 are both significantly correlated with time in the subsurface layer.

[Figure]

**Question 22.** Figure 6: It is very difficult to identify the taxonomic affiliation of even the largest nodes, due to the selected colour-scheme. In addition, considering how common red-green colour-blindness is in the general population, it would be impossible for a large proportion of readers to distinguish between red and green nodes. I therefore recommend that the colour-scheme is reworked for this figure.

**Response:**

We are sorry for the color usage. We have used a better colour scheme to enhance visibility.

[Figure]

**Figure 6: Bacterial Co-occurrence networks for the surface and subsurface layers communities.** Each node represents a bacterial amplicon sequence variant (ASV). The red solid lines represent positive correlations, and the blue solid lines represent negative correlations. Nodes are colored by taxonomy at the phylum level. The subsurface community networks are more complex with a higher positive-to-total correlation ratio.

---

## Editor Decision (ED1)

**Editor report: tc-2021-215**

Dear authors,

Thank you for your thorough response to reviews and amended manuscript. I am happy to recommend publication subject to the final minor amendments listed below.

*Abstract*

The true novelty of this manuscript lies in the study area. Therefore I suggest that rather than the rather bland opening statement, the abstract opens with something in the following format:

'The Tibetan Plateau is important because…. There is little data available….. The snowpack supplies water and potentially nutrients to xxx downstream systems.'

This makes the in situ dynamics on the plateau the subject, not the nutrient export story, which, while fascinating, is not really the central feature of your study.

*Introduction*

I suggest that the introduction also opens with a sentence similar to that suggested above to get your readers interested. Perhaps the paragraph beginning L70 would be better to lead with? I would also recommend highlighting the difference between papers that are on other cryosphere systems (e.g. Arctic glaciers) vs. those more local to your study site. I recommend this strategy to demonstrate that this is a problem that has received insufficient attention in the literature, and thus justify your study.

In the introduction, it would also be worth highlighting when papers are quantifying snowpack exports vs. whole glacier system exports. Again, it might be that there are few snow studies to highlight, in which case: your study is ever more important.

L72: 'Glacier melting increases the discharge of microorganisms and nutrients in meltwater into downstream aquatic ecosystems (Kohler et al., 2020), which substantially impacts the bacterial community and biogeochemical processes (Liu et al., 2021)'.

The way this sentence is structured makes it seem that the Kohler paper is discussing export from Tibetan Plateau systems – it is not. Please make clear that the Liu paper is directly applicable to your study area, whereas the Kohler paper is about potentially similar processes occurring in Arctic systems.

L324: add 'with time' i.e. 'showed a weak increasing trend over time in the surface snow'

L387: suggest replacing 'the present study' with 'this study'

Thank you for your contribution to The Cryosphere.

Dr Liz Bagshaw

---

## Author Response (AR2)

The authors have made some good revisions to the paper. I think there are still some that require change before accepting for publication as detailed below.

**Question 1:** Overall, the English needs a quick check throughout - there are numerous errors that should be caught quite easily.

**Response:**

We thank for this comment. We did a throughout check on the manuscript.

**Intro:**

**Question 2:** I can buy the new rationale of examining processes in recently deposited snow. This is stronger than the previous manuscript. I still don't like reference to only cyanobacteria dominating surface snow. Though you have chosen to only look at bacteria, many eukaryotes are more likely to dominate surface snow. Whilst you have catalogued bacterial diversity here relative to snow chemistry, what if there are eukaryotes present using up/producing the chemical species you are measuring?

**Response:**

We thank for the comment. We tried to clarify this better in the revised version. We did not intend to mean that Cyanobacteria are the dominant microbial group in the surface snow, but Cyanobacteria are more abundant in the surface snow than in subsurface snow. Thus, we have amended the introduction to clarify this and included other taxa that are more abundant in the surface than in the subsurface snow. We have also included the potential impact of algae on nitrogen assimilation in the discussion (algae cannot perform nitrogen fixation).

**Introduction part:**

Original manuscript:

Surface and subsurface typically have distinct bacterial community structures due to the environmental filtering from the vertical profile of temperature, solar radiation

intensity, and nutrients (Xiang et al., 2009; Møller et al., 2013; Carey et al., 2016). For example, Cyanobacteria tend to dominate upper snow layers (0-15 cm) (Carey et al., 2016), while their relative abundance is greatly reduced in the deeper snow layer (Xiang et al., 2009). This is likely due to the lower light intensity in the deeper snow, which favors heterotrophic bacteria such as the Actinobacteria and Firmicutes.

Amended manuscript (Lines 55-65):

Surface and subsurface snow typically harbour distinct bacterial community structures (Xiang et al., 2009; Møller et al., 2013; Carey et al., 2016). For example, algae (chloroplasts), Proteobacteria, Bacteroidetes, and Cyanobacteria were more abundant in surface snow, while Firmicutes and Fusobacteria were more abundant in the deeper snow layer (Møller et al., 2013). A previous study had proposed that nitrogen availability could also be a driver of microbial community structure and function in snow (Larose et al., 2013b), where the $NO_3^-$ and $NH_4^+$ concentrations drove the community composition in Ny-Ålesund snowpack. A dissolved inorganic nitrogen addition experiment also showed a clear community response with the bacterial abundance elevated and genera richness declined in the final time point compared to the initial time point, suggesting potential specialization of heterotrophic communities (Holland et al., 2020).

**Discussion part:**

Amended manuscript (Lines 364-368):

Alternatively, microorganisms may carry out assimilatory nitrate reduction, which is used to incorporate nitrogen into biomolecules (Larose et al., 2013a; Richardson and Watmough, 1999). The assimilatory process is performed by a range of microorganisms including bacteria, algae, yeasts, and fungi (Huth and Liebs, 1988). Thus, further studies on eukaryotes, including algae, may provide a full understanding of the nitrogen consumption mechanisms in subsurface snow.

**Methods:**

**Question 3:** The authors have now removed their TN data, but did not really provide an explanation as to why it was so contradictory to their inorganic nutrient species datasets, i.e. are we happy that the inorganic datasets are reliable, whilst apparently the total nitrogen datasets were not?

**Response:**

We appreciate the reviewer for raising this concern. We have carefully examined standard methods for TN determination and proposed the following possible explanations for the underestimation of TN:

1.  Total nitrogen, nitrate, and ammonium were from independent measurements using different methods. The lower total nitrogen than the sum of nitrate and ammonium could be due to the accumulation of measurement variance. i.e., the measurement variance of TN, nitrate, and ammonium (dissolved inorganic nitrogen, DIN) exceeds the concentration of organic nitrogen. Lee et al. (2005) gave several examples that total dissolved nitrogen could be less than dissolved inorganic nitrogen. A similar conclusion was reported by Sharp et al. (2002) that measurement variance greatly impacts the dissolved organic nitrogen measurement (i.e., the difference between TN and DIN). This is particularly vital for low biomass ecosystems such as surface waters and drinking waters.

2.  The accuracy of measurement may depend on the standards (organic and different forms of inorganic compounds, such as ammonia and nitrate) (Pathak et al., 2015). In the present study, nitrate was used as the reference to calculate total nitrogen, which may underestimate ammonium and organic nitrogen fractions.

3.  The range of the standard curve for total nitrogen quantification was much smaller than the total nitrogen value obtained, and this is most likely to cause the underestimation of total nitrogen and greatly increase the measurement variance.

Based on these reasons, we have decided to remove the total nitrogen results from the manuscript.

Lee, W., Westerhoff, P.J.E.S., and Technology (2005). Dissolved organic nitrogen

measurement using dialysis pretreatment. 39(3), 879-884.

Sharp, J.H., Rinker, K.R., Savidge, K.B., Abell, J., Benaim, J.Y., Bronk, D., et al. (2002). A preliminary methods comparison for measurement of dissolved organic nitrogen in seawater. 78(4), 171-184.

Pathak, B., Al-Omari, A., Wadhawan, T., Higgins, M., and Murthy, S.J.P.o.t.W.E.F. (2015). Analytical errors in the measurement of dissolved organic nitrogen in wastewater effluent. Proceedings of the Water Environment Federation 2015(10), 4214-4225.

**Question 4:** No details on random forest analysis that is now implemented, ie. On what data, and to what end?

**Response:**

We have replaced the random forest analysis with multiple linear regression, please see response below in Q6.

**Results:**

**Question 5:** Whilst the authors have now switched from correlations to mainly regressions, they are still interpreting the outputs incorrectly in my opinion, i.e. line ~225 'The concentration of NO3 and NH4 … increased with time' quotes $R^2$ values of 0.27 and 0.35, respectively. Thus ~ 70% of the variability in NO3 and NH4 was NOT related to the time of sampling….I would not interpret this as a strong indication that NO3 and NH4 increased with time (a quick look at the associated plots supports my conclusions here). The trends are stronger in the subsurface snow but similar issues are found throughout the rest of the results section e.g. info on bacterial community composition through time. I would not recommend publication until this has been amended. Just because you get a 'significant' p-value on a regression does not mean the relationship is 'significant' or even important. You need to look at e.g. $R^2$ values and think about the effect size.

**Response:**

We thank the reviewer for this constructive comment. We have revised the interpretation of the low $R^2$ results in the Results and Discussion sections and clarify that the associations, although significant, were weak.

Original manuscript:

[revised manuscript text omitted]

**Question 6:** Why has random forest analysis been applied here rather than suggested multiple linear regression? Perhaps because MLR does not give good predictive capability on the datasets? This does not seem to provide an explainable test of the relative importance of different factors in reproducing the data. No metrics on the goodness of fit of the RF model are provided in the text.

**Response:**

We appreciate the reviewer for raising this comment. We have performed multiple linear regression to determine the contribution and significance of the environmental characteristics to the alpha diversity. The stepwise AIC method was also performed to select the best model. This result is consistent with the linear regression results, which

identified the concentration of nitrate and ammonium as the significant determinants of bacterial diversity in the subsurface snow layer. We have added this result in the Result section and supplementary as Table S2.

Original manuscript:

This is consistent with the random forest analysis results, which identified the concentrations of $NO_3^-$ and $NH_4^+$ as the significant determinants of bacterial Shannon diversity in the subsurface layer (Supplementary Fig. S8).

Amended manuscript (Lines 241-243):

This is consistent with the multiple linear regression results, which consistently identified the concentrations of $NO_3^-$ and $NH_4^+$ as the significant determinants of bacterial Shannon diversity in the subsurface layer (Supplementary Table S2).

Table S2 Results of multiple linear regression using Akaike's information criterion (AIC), correlating community alpha diversity with environmental variables. Only significant variables were displayed. Best models are in bold.

| | Diversity index | Formula | AIC | $R^2$ | *P* | Explanatory variables |
|---|---|---|---|---|---|---|
| Surface | Shannon | Shannon ~ $NO_3^-$ + $NH_4^+$ + $K^+$ + $SO_4^{2-}$ + DOC + $Na^+$ | -34.69 | 0.28 | *0.14* | $NH_4^+$ (-2.36)* |
| | | Shannon ~ $NH_4^+$ + $K^+$ + $SO_4^{2-}$ + DOC + $Na^+$ | -36.58 | 0.33 | *0.07* | $NH_4^+$ (-2.46)* |
| | | Shannon ~ $NH_4^+$ + $SO_4^{2-}$ + DOC + $Na^+$ | -37.53 | 0.35 | *0.05* | $NH_4^+$ (-2.43)* |
| | | Shannon ~ $NH_4^+$ + $SO_4^{2-}$ + DOC | -38.18 | 0.35 | *0.03* | $NH_4^+$ (-2.28)* |
| | | **Shannon ~ $NH_4^+$ + DOC** | **-38.44** | **0.33** | ***0.02*** | **DOC (3.20)**\*\* |
| | Chao1 | Chao1 ~ $NO_3^-$ + $NH_4^+$ + $K^+$ + $SO_4^{2-}$ + DOC + $Na^+$ | 194.21 | 0.07 | *0.36* | |
| | | Chao1 ~ $NO_3^-$ + $NH_4^+$ + $K^+$ + $SO_4^{2-}$ + DOC | 192.22 | 0.15 | *0.23* | |

| | | | | | | |
|---|---|---|---|---|---|---|
| | | Chao1 ~ $NO_3^-$ + $NH_4^+$ + $SO_4^{2-}$ + DOC | 190.59 | 0.20 | *0.15* | |
| | | Chao1 ~ $NO_3^-$ + $SO_4^{2-}$ + DOC | 189.54 | 0.22 | *0.1* | |
| | | **Chao1 ~ $NO_3^-$ + $SO_4^{2-}$** | **188.72** | **0.22** | ***0.06*** | **$SO_4^2$ (2,54)\*** |
| Subsurface | Shannon | Shannon ~ $NO_3^-$ + $NH_4^+$ + $K^+$ + $SO_4^{2-}$ + DOC + $Na^+$ | -25.59 | 0.61 | *0.008* | $NO_3^-$ (3.79)\*\*, $NH_4^+$ (-2.54)\* |
| | | **Shannon ~ $NO_3^-$ + $NH_4^+$ + $SO_4^{2-}$ + DOC + $Na^+$** | **-26.91** | **0.63** | ***0.003*** | **$NO_3^-$ (3.98)\*\*, $NH_4^+$ (-2.76)\*, $SO_4^2$ (-2.20)\*** |
| | Chao1 | Chao1 ~ $NO_3^-$ + $NH_4^+$ + $K^+$ + $SO_4^{2-}$ + DOC + $Na^+$ | 183.77 | 0.73 | *0.001* | $NO_3^-$ (5.02)\*\*\*, $SO_4^{2-}$ (-4.52)\*\*\*, $Na^+$ (4.34)\*\* |
| | | Chao1 ~ $NO_3^-$ + $NH_4^+$ + $K^+$ + $SO_4^{2-}$ + $Na^+$ | 181.77 | 0.76 | *<0.001* | $NO_3^-$ (5.25)\*\*\*, $NH_4^+$ (-2.41)\*, $SO_4^2$ (-5.20)\*\*\*, $Na^+$ (5.22)\*\*\* |
| | | **Chao1 ~ $NO_3^-$ + $NH_4^+$ + $SO_4^{2-}$ + $Na^+$** | **181.25** | **0.75** | ***<0.001*** | **$NO_3^-$ (5.40)\*\*\*, $NH_4^+$ (-2,67)\*, $SO_4^2$ (-5.48)\*\*\*, $Na^+$ (5.40)\*\*\*** |

\* $P < 0.05$, \*\* $P < 0.01$, \*\*\* $P < 0.001$.

**Discussion:**

**Question 7:** The same caveats as listed for presentation of the results section apply to the degree of interpretation apparent in the discussion section.

**Response:**

We thank for the suggestion. We have changed the interpretation of nitrogen changes in the surface snow as unchanged or slightly increased, which indicates the microbiome is not subjected to nitrogen-limitation, rather than an accumulation process.

We amended the abstract, discussion, and conclusion.

**Abstract part:**

Amended manuscript (Lines 20-22):

Therefore, we suggest that the surface snow is not nitrogen-limited, while the subsurface snow is associated with nitrogen consumption processes and nitrogen limited.

**Discussion part:**

Amended manuscript:

Lines 292-294:

Bacterial richness and diversity exhibited little change throughout the nine days in the surface snow layer, while they exhibited a reduction trend in the subsurface snow layer (Fig. 3b).

Lines 306-308:

In comparison, the surface layer is unlikely to be subjected to nitrogen-limitation and the nitrogen in the surface snow slightly increased.

Lines 311-313:

The bacterial community structure also exhibited temporal changes in the subsurface layer. Furthermore, associations between nitrogen and the microbial community

structure were observed to a certain degree (Table 1 and Fig. 5), again indicating some level of environmental filtering (Kim et al., 2016).

Lines 322-324:

Our results suggest that both bacteria and snow physiochemical properties experience changes across the nine days during the snow deposition period in the Tibetan glacier investigated here, and those changes were more stronger in the subsurface layer than in the surface layer.

Lines 335-337:

Both ammonium and nitrate concentrations showed a weak increasing trend in the surface snow (Fig. 1). The weak increase in ammonium could be explained by biogenic emissions due to local plant and animal sources (Filippa et al., 2010)

**Conclusion part:**

Amended manuscript (Lines 411-412):

Inorganic nitrogen was unchanged or slightly increased in the surface snow, while it decreased in subsurface snow.

**Question 8:** Back of the envelope calculations of potential N deposition are obviously likely to be highly inaccurate, so using this to justify N-fixation is tenuous at best. Better to just outline the possible causes of N increase (if indeed there was N increase) of deposition versus biological processes. You are inferring a lot here - as was one of the main criticisms of the previous discussion.

**Response:**

We thank for the constructive suggestion. We have deleted the inferring part about the potential N deposition and revised this part to only discuss the possible nitrogen source in the surface snow layer.

Both ammonium and nitrate concentrations showed a weak increasing trend in the surface snow (Fig. 1). The weak increase in ammonium could be explained by biogenic emissions due to local plant and animal sources (Filippa et al., 2010), while the increase in nitrate has been largely attributed to atmospheric deposition (Björkman et al., 2014). Nitrogen deposition occurs at a rate of 282 kg N km$^{-2}$ yr$^{-1}$ in the region of our investigation (Lü and Tian, 2007), which equals to 0.19 mg N for the 0.5 m $\times$ 0.5 m area sampled each day (assuming nitrogen deposition occurred evenly across the year). Another potential source of nitrogen input could be nitrogen fixation process (Telling et al., 2011). Bacteria are the only microorganisms that are capable of fixing atmospheric nitrogen (Bernhard, 2010). Potential nitrogen input from microbial processes is supported by the increase in the nitrogen-fixing Cyanobacteria (Supplementary Fig. S6) and *nifH* gene (Supplementary Fig. S11). Cyanobacteria are known as free-living phototrophs capable of nitrogen fixation, especially in extreme environments (Chrismas et al., 2018; Makhalanyane et al., 2015; Levy-Booth et al., 2014). For example, Cyanobacteria were found as the main group of potential nitrogen fixers determined by quantitative PCR with three sets of specific *nifH* primers on the surface of the Greenland Ice Sheet (Telling et al., 2012). The nitrogen fixation rate was not quantified in the present study, but the present study suggests that microbial nitrogen fixation could be an overlooked source of nitrogen in Tibetan glacier snow. Further transcriptomic and nitrogen-isotope analyses may provide additional evidence on the microbial activity in nitrogen fixation.

**Question 9:** Nitrogen use in sub-surface snow; again, I would simply outline potential pathways of N use rather than trying to make a definitive conclusion here based on inferred datasets. You can't really claim with certainty the denitrification story pushed here.

**Response:**

We thank for the comment. We have rewritten this part to discuss the potential pathways

of N use in the subsurface layer.

In contrast with the surface layer, nitrogen concentrations (nitrate and ammonium) significantly decreased in the subsurface snow with time (Fig. 1). A possible explanation for this might be the microbial utilization and photochemical degradation of nitrogen compounds (Björkman et al., 2014). The microbial processes, i.e. nitrate reduction and denitrification process, are evidenced by the increase of *narG* gene (Supplementary Fig. S11) (Telling et al., 2011; Zhang et al., 2020). Alternatively, microorganisms may carry out assimilatory nitrate reduction, which is used to incorporate nitrogen into biomolecules (Larose et al., 2013a; Richardson and Watmough, 1999). The assimilatory process is performed by a range of microorganisms including bacteria, algae, yeasts, and fungi (Huth and Liebs, 1988). Thus, further studies on eukaryotes, including algae, may provide a full understanding of the nitrogen consumption mechanisms in subsurface snow. The denitrification process converts nitrate to $N_2$ and generates nitrite, nitric oxide (NO), and nitrous oxide ($N_2O$) intermediates (Kuypers et al., 2018). A previous study detected microbial specific phylogenetic probes that targeted genera whose members are able to carry out denitrification reactions such as Roseomonas in a snowpack of Spitsbergen Island of Svalbard, Norway (Larose et al., 2013a). Amoroso et al. (2010) also proposed that denitrification can explain the microbial isotopic signature observed in winter snow at Ny-Alesund. Although the oxygen level in the subsurface snow was not measured, the occurrence of anaerobic denitrification reactions in subsurface snow has been reported in Arctic snowpacks (Larose et al., 2013a). Lastly, photochemical degradation of nitrogen compounds is the most well-known nitrogen degradation pathway, and the release of both NO and $NO_x$ by $NO_3^-$ photolysis on natural snow has been reported in European High Arctic snowpack (Amoroso et al., 2010; Beine et al., 2003). In a snow reactive nitrogen oxides ($NO_y$) survey in Greenland, $NO_y$ flux was reported to exit snow in 52 out of 112 measurements (Dibb et al., 1998).

**Question 10:** I would amend the conclusion as well based on comments above.

**Response:**

We thank the reviewer for the constructive suggestion. We rewrite the conclusion section based on the above modifications.

Amended manuscript (Lines 410-422):

Our results showed the dynamics of nitrogen and bacterial community in supraglacial snow over nine days. Inorganic nitrogen was unchanged or slightly increased in the surface snow, while it decreased in subsurface snow. Due to atmospheric nitrogen deposition and potentially bacterial nitrogen fixation activities, nitrogen limitation is unlikely to occur in the surface snow. In contrast, nitrogen consumption was inferred in the subsurface snow. Nitrogen is traditionally recognized to be released from the supraglacial environment due to photolysis, whereas the present study hints that nitrogen assimilation and denitrification could be alternative routes. Therefore, the increased nitrogen deposition due to anthropogenic activities may enhance the nitrogen consumption in the subsurface snow, which reduces the impact of increased nitrogen discharge on downstream glacier-fed rivers. In summary, our results provide a new perspective of the nutrients and bacterial community dynamics in supraglacial snow of the Tibetan Plateau. Further studies based on metagenome and metatranscriptome can enhance the understanding of bacterial functions.

**Figures**

**Question 11:** Eyeballing the revised figures again shows there is no real increase in NO3 of NH4 through time above the variability evident in the data. 95% CIs should be shown on regressions.

**Response:**

We thank for the comment. We added the 95% CIs on regressions. Please see Figures and Supplementary.

[Figure]

**Fig. 1 The pattern of environmental factors changes in the surface and subsurface snow layers.**
(a) Environmental factor comparisons in the surface and subsurface snow layers. Each dot represents an individual sample. Significantly higher concentrations of $NO_3^-$, $NH_4^+$, $K^+$, and $SO_4^{2-}$ were observed in the subsurface layer based on Wilcoxon rank-sum test. (b) Temporal changes of environmental factors in the surface and subsurface layers. The solid and dashed lines indicate significant and non-significant temporal changes, respectively. The concentration of $NO_3^-$ and $NH_4^+$ in the surface layer significantly increased with time while the concentration of $NO_3^-$, and $NH_4^+$, in the subsurface layer, significantly decreased with time. Significance is based on linear regression. Grey shading indicates the 95% confidence interval of regression.

[Figure]

**Fig. 3 Bacterial alpha diversity in snow layers.** (a) Bacterial alpha diversity comparison between the surface and subsurface layers. Each dot represents an individual sample. For both Shannon and Chao1 indices, no significant difference was observed between the surface and subsurface snow layers. Comparison is based on Wilcoxon rank-sum test. (b) Temporal changes of the alpha diversity indices in the surface and subsurface snow layers. For the surface layer, no significant correlation was observed, while both Shannon and Chao1 showed a significantly reduction with time in the subsurface layer. Significance is based on linear regression. Grey shading indicates the 95% confidence interval of regression.

[Figure]

**Fig. 4 The influence of environmental factors on bacterial diversity.** Correlations of Shannon (a, c) and Chao1 (b, d) diversity indices with environmental factors in the surface and subsurface layers. Each dot represents an individual sample. The solid and dashed lines indicate significant and nonsignificant changes respectively. Significance is based on linear regression. Grey shading indicates the 95% confidence interval of regression.

[Figure]

**Fig. 5 Principal coordinate analysis (PCoA) of microbial communities in the surface and subsurface snow.** (a) Bray-Curtis distance-based PCoA ordination plot. The microbial community structures of the surface and subsurface snows are significantly different (PERMANOVA, $P <$ 0.001). (b) Pairwise regression analysis between PCoA scores and sampling time. The solid and dashed lines indicate significant and insignificant changes (based on linear regression), respectively. The PCoA1 scores for the bacterial community in the surface layer exhibit no significant correlation with time, while the PCoA2 scores significantly correlated with time. The PCoA1 and PCoA2 are both significantly correlated with time in the subsurface layer. Grey shading indicates the 95% confidence interval of regression.

---

## Author Response (AR3)

We sincerely thank the editor for the constructive comments provided. We have carefully amended our manuscript accordingly, please see below.

**Abstract**

**Question 1:** The true novelty of this manuscript lies in the study area. Therefore I suggest that rather than the rather bland opening statement, the abstract opens with something in the following format: 'The Tibetan Plateau is important because…. There is little data available….. The snowpack supplies water and potentially nutrients to xxx downstream systems.' This makes the in situ dynamics on the plateau the subject, not the nutrient export story, which, while fascinating, is not really the central feature of your study.

**Response:**

We thank the editor for this comment. We have modified the beginning of the abstract to emphasize the value of the present study to Tibetan glaciers and the downstream ecosystems.

Amended manuscript:

The Tibetan Plateau harbours the largest number of glaciers outside the polar regions, which are the source of several major rivers in Asia. These glaciers are also major sources of nutrients for downstream ecosystems, while there is little data available on the nutrient transformation processes on the glacier surface. Here, we monitored the carbon and nitrogen concentration changes in a snowpit following a snowfall in the Dunde Glacier of the Tibetan Plateau. The association of carbon and nitrogen changes with bacterial community dynamics was investigated in the surface and subsurface snow (depth at 0-15 and 15-30 cm, respectively) during a nine-day period.

**Introduction**

**Question 2:** I suggest that the introduction also opens with a sentence similar to that suggested above to get your readers interested. Perhaps the paragraph beginning L70 would be better to lead with? I would also recommend highlighting the difference between papers that are on other cryosphere systems (e.g. Arctic glaciers) vs. those more local to your study site. I recommend this strategy to demonstrate that this is a problem that has received insufficient attention in the literature, and thus justify your study.

**Response:**

We thank the editor for this suggestion. We have reconstructed our introduction and it

now opens with the importance of Tibetan glaciers. We also presented the available literature on the microbial and nutrient dynamics in other cryosphere ecosystems, including the glaciers of the Arctic, Antarctic, and the Alps. We then emphasize the lack of similar knowledge in Tibetan glaciers, which is the centre of the present study.

Amended manuscript:

The Tibetan Plateau (TP) is the world's third-largest ice reservoir, after those in Antarctica and Greenland (Qiu, 2012). These glaciers are the source of several large rivers in Asia, such as the Yellow, Yangtze, Mekong, Salween, Brahmaputra, and Indus rivers (Immerzeel et al., 2010). Glaciers are major sources of nutrients (carbon and nitrogen) for downstream ecosystems (Singer et al., 2012; Hood et al., 2015; Liu et al., 2021). It has been estimated that 80 gigagram of dissolved organic carbon and 27-43 gigagram of nitrogen are exported from the Greenland Ice Sheet (Bhatia et al., 2013; Wadham et al., 2016). These nutrients are subjected to complex accumulation and transformation processes in the glacier snow before being released into downstream ecosystems, and microorganisms are the drivers of these processes (Anesio and Laybourn-Parry, 2012; Hell et al., 2013; Hodson et al., 2008). Several studies on snowpacks revealed vital knowledge of the nutrient and microbial community dynamics in the Arctic (Hell et al., 2013; Larose et al., 2013a; Larose et al., 2013b; Maccario et al., 2014; Maccario et al., 2019), Antarctic (Antony et al., 2016), and Alps (Lazzaro et al., 2015). However, such knowledge is rarely available in the Tibetan Plateau, constraining our understanding of the nutrient accumulation, transformation, and release processes, which is urgently needed under the enhanced warming and glacier retreat in the Tibetan Plateau.

**Question 3:** In the introduction, it would also be worth highlighting when papers are quantifying snowpack exports vs. whole glacier system exports. Again, it might be that there are few snow studies to highlight, in which case: your study is ever more important.

**Response:**

We appreciate the editor for this comment. We have added additional sentences on the quantity of carbon and nitrogen exported from glaciers, then used this as a background for the value of understanding the accumulation and transformation processes of these nutrients.

Glaciers are major sources of nutrients (carbon and nitrogen) for downstream ecosystems (Singer et al., 2012; Hood et al., 2015; Liu et al., 2021). It has been estimated that 80 gigagram of dissolved organic carbon and 27-43 gigagram of nitrogen are exported from the Greenland Ice Sheet (Bhatia et al., 2013; Wadham et al., 2016).

These nutrients are subjected to complex accumulation and transformation processes in the glacier snow before being released into downstream ecosystems, and microorganisms are the drivers of these processes (Anesio and Laybourn-Parry, 2012; Hell et al., 2013; Hodson et al., 2008). Several studies on snowpacks revealed vital knowledge of the nutrient and microbial community dynamics in the Arctic (Hell et al., 2013; Larose et al., 2013a; Larose et al., 2013b; Maccario et al., 2014; Maccario et al., 2019), Antarctic (Antony et al., 2016), and Alps (Lazzaro et al., 2015). However, such knowledge is rarely available in the Tibetan Plateau, constraining our understanding of the nutrient accumulation, transformation, and release processes, which is urgently needed under the enhanced warming and glacier retreat in the Tibetan Plateau.

The following sentences describe the knowledge status quo on the carbon and nitrogen dynamics in glacier surface across a long period of time (such as the ablation period or across a whole season), and then emphasize the lack of knowledge on nutrient dynamics in snowpit (i.e., from single precipitation across a short temporal scale), which is more relevant to microbial transformation processes.

Autochthonous (microbial origin) and allochthonous (wet and dry atmospheric depositions) are the major sources of nutrients on supraglacial snow, and the contribution of allochthonous sources was much greater in Arctic glaciers (Larose et al., 2013a). Microorganisms are highly involved in the transformation of both autochthonous and allochthonous nutrients. Several studies investigated the dynamics of nutrient and bacterial changes in supraglacial snow during the ablation period. Larose et al. (2013a) revealed that the form of nitrogen varied as a function of time in supraglacial snow during a two-month field study in Svalbard, and fluctuations in microbial community structure were reported with the relative abundance of fungi and bacteria (such as Bacteroidetes and Proteobacteria) increased and decreased, respectively. Seasonal shifts in snowpack bacterial communities were reported in the mountain snow in Japan, where rapid microbial growth was observed with increasing snow temperature and meltwater content (Segawa et al., 2005). However, the results of these studies are likely the consequence of several precipitation events due to the long study period. During precipitation, a new snow layer forms above the previous ones, which is responsible for the stratified snowpack structure. These different snow layers have distinct physical and chemical characteristics and their age also differs substantially (Lazzaro et al., 2015). Thus, while the microbial process across the aged snowpack can be complex, focusing on supraglacial snow from a single snowfall event could provide unique insights into the bacterial and nutrient dynamics. For instance, Hell et al. (2013) reported bacterial community structure changes during the ablation period across five days in the high Arctic, but the bacterial and nutrient dynamics during the snow accumulation period remain elusive.

**Question 4:** L72: 'Glacier melting increases the discharge of microorganisms and nutrients in meltwater into downstream aquatic ecosystems (Kohler et al., 2020), which substantially impacts the bacterial community and biogeochemical processes (Liu et al., 2021)'. The way this sentence is structured makes it seem that the Kohler paper is discussing export from Tibetan Plateau systems – it is not. Please make clear that the Liu paper is directly applicable to your study area, whereas the Kohler paper is about potentially similar processes occurring in Arctic systems.

**Response:**

We appreciate the editor for raising this concern. Due to the modification on the introduction structure, this sentence has been removed.

**Question 5:** L324: add 'with time' i.e. 'showed a weak increasing trend over time in the surface snow'

**Response:**

We have revised the sentence accordingly.

Original manuscript:

Both ammonium and nitrate concentrations showed a weak increasing trend in the surface snow (Fig. 1).

Amended manuscript:

Both ammonium and nitrate concentrations showed a weak increasing trend with time in the surface snow (Fig. 1).

**Question 6:** L387: suggest replacing 'the present study' with 'this study'

**Response:**

We have revised the sentence accordingly.

Original manuscript:

Nitrogen is traditionally recognized to be released from the supraglacial environment due to photolysis, whereas the present study hints that nitrogen assimilation and denitrification could be alternative routes.

Amended manuscript:

Nitrogen is traditionally recognized to be released from the supraglacial environment due to photolysis, whereas this study hints that nitrogen assimilation and denitrification could be alternative routes.